# MARTINGALE POSTERIOR NEURAL PROCESSES

**Hyungi Lee[1], Eunggu Yun[1], Giung Nam[1], Edwin Fong[2], Juho Lee[1,3]**

[1]KAIST, [2]Novo Nordisk, [3]AITRICS
[1]{lhk2708, eunggu.yun, giung, juholee}@kaist.ac.kr, [2]chef@novonordisk.com

## ABSTRACT

A Neural Process (NP) estimates a stochastic process implicitly defined with neural networks given a stream of data, rather than pre-specifying priors already known, such as Gaussian processes. An ideal NP would learn everything from data without any inductive biases, but in practice, we often restrict the class of stochastic processes for the ease of estimation. One such restriction is the use of a finite-dimensional latent variable accounting for the uncertainty in the functions drawn from NPs. Some recent works show that this can be improved with more "data-driven" source of uncertainty such as bootstrapping. In this work, we take a different approach based on the martingale posterior, a recently developed alternative to Bayesian inference. For the martingale posterior, instead of specifying prior-likelihood pairs, a predictive distribution for future data is specified. Under specific conditions on the predictive distribution, it can be shown that the uncertainty in the generated future data actually corresponds to the uncertainty of the implicitly defined Bayesian posteriors. Based on this result, instead of assuming any form of the latent variables, we equip a NP with a predictive distribution implicitly defined with neural networks and use the corresponding martingale posteriors as the source of uncertainty. The resulting model, which we name as Martingale Posterior Neural Process (MPNP), is demonstrated to outperform baselines on various tasks.

## 1 INTRODUCTION

A Neural Process (NP) (Garnelo et al., 2018a;b) meta-learns a stochastic process describing the relationship between inputs and outputs in a given data stream, where each task in the data stream consists of a meta-training set of input-output pairs and also a meta-validation set. The NP then defines an implicit stochastic process whose functional form is determined by a neural network taking the meta-training set as an input, and the parameters of the neural network are optimized to maximize the predictive likelihood for the meta-validation set. This approach is philosophically different from the traditional learning pipeline where one would first elicit a stochastic process from the known class of models (e.g., Gaussian Processes (GPs)) and hope that it describes the data well. An ideal NP would assume minimal inductive biases and learn as much as possible from the data. In this regard, NPs can be framed as a "data-driven" way of choosing proper stochastic processes.

An important design choice for a NP model is how to capture the uncertainty in the random functions drawn from stochastic processes. When mapping the meta-training set into a function, one might employ a deterministic mapping as in Garnelo et al. (2018a). However, it is more natural to assume that there may be multiple plausible functions that might have generated the given data, and thus encode the functional (epistemic) uncertainty as a part of the NP model. Garnelo et al. (2018b) later proposed to map the meta-training set into a fixed dimensional *global latent variable* with a Gaussian posterior approximation. While this improves upon the vanilla model without such a latent variable (Le et al., 2018), expressing the functional uncertainty only through the Gaussian approximated latent variable has been reported to be a bottleneck (Louizos et al., 2019). To this end, Lee et al. (2020) and Lee et al. (2022) propose to apply bootstrap to the meta-training set to use the uncertainty arising from the population distribution as a source for the functional uncertainty.

In this paper, we take a rather different approach to define the functional uncertainty for NPs. Specifically, we utilize the martingale posterior distribution (Fong et al., 2021), a recently developed alternative to conventional Bayesian inference. In the martingale posterior, instead of eliciting a

likelihood-prior pair and inferring the Bayesian posterior, we elicit a joint predictive distribution on future data given observed data. Under suitable conditions on such a predictive distribution, it can be shown that the uncertainty due to the generated future data indeed corresponds to the uncertainty of the Bayesian posterior. Following this, we endow a NP with a joint predictive distribution defined through neural networks and derive the functional uncertainty as the uncertainty arising when mapping the randomly generated future data to the functions. Compared to the previous approaches of either explicitly positing a finite-dimensional variable encoding the functional uncertainty or deriving it from a population distribution, our method makes minimal assumptions about the predictive distribution and gives more freedom to the model to choose the proper form of uncertainty solely from the data. Due to the theory of martingale posteriors, our model guarantees the existence of the martingale posterior corresponding to the valid Bayesian posterior of an implicitly defined parameter. Furthermore, working in the space of future observations allows us to incorporate the latent functional uncertainty path with deterministic path in a more natural manner.

We name our extension of NPs with the joint predictive generative models as the Martingale Posterior Neural Process (MPNP). Throughout the paper, we propose an efficient neural network architecture for the generative model that is easy to implement, flexible, and yet guarantees the existence of the martingale posterior. We also propose a training scheme to stably learn the parameters of MPNPs. Using various synthetic and real-world regression tasks, we demonstrate that MPNP significantly outperforms the previous NP variants in terms of predictive performance.

## 2 BACKGROUND

### 2.1 SETTINGS AND NOTATIONS

Let $\mathcal{X} = \mathbb{R}^{d_{\text{in}}}$ be an input space and $\mathcal{Y} = \mathbb{R}^{d_{\text{out}}}$ be an output space. We are given a set of *tasks* drawn from an (unknown) task distribution, $\tau_1, \tau_2, \ldots \overset{\text{i.i.d.}}{\sim} p_{\text{task}}(\tau)$. A task $\tau$ consists of a dataset $Z$ and an index set $c$, where $Z = \{z_i\}_{i=1}^n$ with each $z_i = (x_i, y_i) \in \mathcal{X} \times \mathcal{Y}$ is a pair of an input and an output. We assume $Z$ are i.i.d. conditioned on some function $f$. The index set $c \subsetneq [n]$ where $[n] := \{1, \ldots, n\}$ defines the *context set* $Z_c = \{z_i\}_{i \in c}$. The *target set* $Z_t$ is defined similarly with the index $t := [n] \setminus c$.

### 2.2 NEURAL PROCESS FAMILIES

Our goal is to train a class of random functions $f : \mathcal{X} \to \mathcal{Y}$ that can effectively describe the relationship between inputs and outputs included in a set of tasks. Viewing this as a meta-learning problem, for each task $\tau$, we can treat the context $Z_c$ as a meta-train set and target $Z_t$ as a meta-validation set. We wish to meta-learn a mapping from the context $Z_c$ to a random function $f$ that recovers the given context $Z_c$ (minimizing meta-training error) and predicts $Z_t$ well (minimizing meta-validation error). Instead of directly estimating the infinite-dimensional $f$, we learn a mapping from $Z_c$ to a predictive distribution for finite-dimensional observations,

$$p(Y|X, Z_c) = \int \left[ \prod_{i \in c} p(y_i|f, x_i) \prod_{i \in t} p(y_i|f, x_i) \right] p(f|Z_c) \mathrm{d}f, \tag{1}$$

where we are assuming the outputs $Y$ are independent given $f$ and $X$. We further restrict ourselves to simple heteroscedastic Gaussian measurement noises,

$$p(y|f, x) = \mathcal{N}(y|\mu_\theta(x), \sigma_\theta^2(x) I_{d_{\text{out}}}), \tag{2}$$

where $\mu_\theta : \mathcal{X} \to \mathcal{Y}$ and $\sigma_\theta^2 : \mathcal{X} \to \mathbb{R}_+$ map an input to a mean function value and corresponding variance, respectively. $\theta \in \mathbb{R}^h$ is a parameter indexing the function $f$, and thus the above predictive distribution can be written as

$$p(Y|X, Z_c) = \int \left[ \prod_{i \in [n]} \mathcal{N}(y_i|\mu_\theta(x_i), \sigma_\theta^2(x_i) I_{d_{\text{out}}}) \right] p(\theta|Z_c) \mathrm{d}\theta. \tag{3}$$

A NP is a parametric model which constructs a mapping from $Z_c$ to $\theta$ as a neural network. The simplest version, Conditional Neural Process (CNP) (Garnelo et al., 2018a), assumes a deterministic mapping from $Z_c$ to $\theta$ as

$$p(\theta|Z_c) = \delta_{r_c}(\theta), \quad r_c = f_{\text{enc}}(Z_c; \phi_{\text{enc}}), \tag{4}$$

where $\delta_a(x)$ is the Dirac delta function (which gives zero if $x \neq a$ and $\int \delta_a(x) \mathrm{d}x = 1$) and $f_{\mathrm{enc}}$ is a *permutation-invariant* neural network taking sets as inputs (Zaheer et al., 2017), parameterized by $\phi_{\mathrm{enc}}$. Given a summary $\theta = r_c$ of a context $Z_c$, the CNP models the mean and variance functions $(\mu, \sigma^2)$ as

$$(\mu_\theta(x), \log \sigma_\theta(x)) = f_{\mathrm{dec}}(x, r_c; \phi_{\mathrm{dec}}), \tag{5}$$

where $f_{\mathrm{dec}}$ is a feed-forward neural network parameterized by $\phi_{\mathrm{dec}}$. Here the parameters $(\phi_{\mathrm{enc}}, \phi_{\mathrm{dec}})$ are optimized to maximize the expected predictive likelihood over tasks, $\mathbb{E}_\tau[\log p(Y|X, Z_c)]$.

Note that in the CNP, the mapping from $Z_c$ to $\theta$ is deterministic, so it does not consider *functional uncertainty* or epistemic (model) uncertainty. To resolve this, Garnelo et al. (2018b) proposed NP which learns a mapping from an arbitrary subset $Z' \subseteq Z$ to a variational posterior $q(\theta|Z')$ approximating $p(\theta|Z')$ under an implicitly defined prior $p(\theta)$:

$$(m_{Z'}, \log s_{Z'}) = f_{\mathrm{enc}}(Z'; \phi_{\mathrm{enc}}), \quad p(\theta|Z') \approx q(\theta|Z') := \mathcal{N}(\theta|m_{Z'}, s_{Z'}^2 I_h). \tag{6}$$

With $f_{\mathrm{enc}}$, the Evidence Lower BOund (ELBO) for the predictive likelihood is written as

$$\log p(Y|X, Z_c) \geq \sum_{i \in [n]} \mathbb{E}_{q(\theta|Z)}[\log \mathcal{N}(y_i|\mu_\theta(x_i), \sigma_\theta^2(x_i) I_{d_{\mathrm{out}}})] - D_{\mathrm{KL}}[q(\theta|Z) \| p(\theta|Z_c)]$$

$$\approx \sum_{i \in [n]} \mathbb{E}_{q(\theta|Z)}[\log \mathcal{N}(y_i|\mu_\theta(x_i), \sigma_\theta^2(x_i) I_{d_{\mathrm{out}}})] - D_{\mathrm{KL}}[q(\theta|Z) \| q(\theta|Z_c)]. \tag{7}$$

An apparent limitation of the NP is that it assumes a uni-modal Gaussian distribution as an approximate posterior for $q(\theta|Z_c)$. Aside from the limited flexibility, it does not fit the motivation of NPs trying to learn as much as possible in a data-driven manner, as pre-specified parametric families are used.

There have been several improvements over the vanilla CNPs and NPs, either by introducing attention mechanism (Vaswani et al., 2017) for $f_{\mathrm{enc}}$ and $f_{\mathrm{dec}}$ (Kim et al., 2018), or using advanced functional uncertainty modeling (Lee et al., 2020; Lee et al., 2022). We provide a detailed review of the architectures for such variants in Appendix A. Throughout the paper, we will refer to this class of models as Neural Process Family (NPF).

## 2.3 MARTINGALE POSTERIOR DISTRIBUTIONS

The martingale posterior distribution (Fong et al., 2021) is a recent generalization of Bayesian inference which reframes posterior uncertainty on parameters as predictive uncertainty on the unseen population conditional on the observed data. Given observed samples $Z = \{z_i\}_{i=1}^n$ i.i.d. from the sampling density $p_0$, one can define the parameter of interest as a functional of $p_0$, that is

$$\theta_0 = \theta(p_0) = \arg\min_\theta \int \ell(z, \theta) \, p_0(dz),$$

where $\ell$ is a loss function. For example, $\ell(z, \theta) = (z - \theta)^2$ would return $\theta_0$ as the mean, and $\ell(z, \theta) = -\log p(z \mid \theta)$ would return the KL minimizing parameter between $p(\cdot \mid \theta)$ and $p_0$.

The next step of the martingale posterior is to construct a *joint* predictive density on $Z' = \{z_i\}_{i=n+1}^N$ for some large $N$, which we write as $p(Z' \mid Z)$. In a similar fashion to a bootstrap, one can imagine drawing $Z' \sim p(Z' \mid Z)$, then computing $\theta(g_N)$ where $g_N(z) = \frac{1}{N} \sum_{i=1}^N \delta_{z_i}(z)$. The predictive uncertainty in $Z'$ induces uncertainty in $\theta(g_N)$ conditional on $Z$. The key connection is that if $p(Z' \mid Z)$ is the Bayesian joint posterior predictive density, and $\ell = -\log p(z \mid \theta)$, then $\theta(g_N)$ is distributed according to the Bayesian posterior $\pi(\theta \mid Z)$ as $N \to \infty$, under weak conditions. In other words, posterior uncertainty in $\theta$ is equivalent to predictive uncertainty in $\{z_i\}_{i=n+1}^\infty$.

Fong et al. (2021) specify more general $p(Z' \mid Z)$ directly beyond the Bayesian posterior predictive, and define the (finite) martingale posterior as $\pi_N(\theta \in A \mid Z) = \int \mathbb{1}(\theta(g_N) \in A) \, p(dZ' \mid Z)$. In particular, the joint predictive density can be factorized into a sequence of 1-step-ahead predictives, $p(Z' \mid Z) = \prod_{i=n+1}^N p(z_i \mid z_{1:i-1})$, and the sequence $\{p(z_i \mid z_{1:i-1})\}_{n+1}^N$ is elicited directly, removing the need for the likelihood and prior. Hyperparameters for the sequence of predictive distributions can be fitted in a data-driven way by maximizing

$$\log p(Z) = \sum_{i=1}^n \log p(z_i \mid z_{1:i-1}),$$

which is analogous to the log marginal likelihood. Fong et al. (2021) requires the sequence of predictives to be conditionally identically distributed (c.i.d.), which is a martingale condition on the sequence of predictives that ensures $g_N$ exists almost surely. The Bayesian posterior predictive density is a special case, as exchangeability of $p(Z' \mid Z)$ implies the sequence of predictives is c.i.d. In fact, De Finetti's theorem (De Finetti, 1937) guarantees that any exchangeable joint density implies an underlying likelihood-prior form, but specifying the predictive density directly can be advantageous. It allows for easier computation, as we no longer require posterior approximations, and it also widens the class of available nonparametric predictives which we will see shortly.

## 2.4 EXCHANGEABLE GENERATIVE MODELS

To construct a martingale posterior, we can either specify a sequence of one-step predictive distributions or the joint predictive density distribution directly, as long as the c.i.d. condition is satisfied. Here, we opt to specify an exchangeable $p(Z' \mid Z)$ directly, which then implies the required c.i.d. predictives. We now briefly review exchangeable generative models which can be used to specify the exchangeable joint predictive. For a set of random variables $Z = \{z_i\}_{i=1}^n$ with each $z_i \in \mathcal{Z} = \mathbb{R}^d$, we say the joint distribution $p(Z)$ is *exchangeable* if it is invariant to the arbitrary permutation of the indices, that is, $p(Z) = p(\pi \cdot Z)$ for any permutation $\pi$ of $[n]$. A simple way to construct such exchangeable random variables is to use a *permutation-equivariant mapping*. A mapping $\mathbf{f} : \mathcal{Z}^n \to \mathcal{Z}^n$ is permutation equivariant if $\mathbf{f}(\pi \cdot Z) = \pi \cdot \mathbf{f}(Z)$ for any $\pi$. Given $\mathbf{f}$, we can first generate i.i.d. random variables and apply $\mathbf{f}$ to construct a potentially correlated but exchangeable set of random variables $Z$ as follows:

$$\mathcal{E} := \{\varepsilon_i\}_{i=1}^n \overset{\text{i.i.d.}}{\sim} p_0, \quad Z = \mathbf{f}(\mathcal{E}). \tag{8}$$

For $\mathbf{f}$, we employ the modules introduced in Lee et al. (2019). Specifically, we use a permutation equivariant module called Induced Self-Attention Block (ISAB). An ISAB mixes input sets through a learnable set of parameters called *inducing points* via Multihead Attention Blocks (MABs) (Vaswani et al., 2017; Lee et al., 2019).

$$\text{ISAB}(\mathcal{E}) = \text{MAB}(\mathcal{E}, H) \in \mathbb{R}^{n \times d} \text{ where } H = \text{MAB}(I, \mathcal{E}) \in \mathbb{R}^{m \times d}. \tag{9}$$

Here, $I \in \mathbb{R}^{m \times d}$ is a set of $m$ inducing points and $\text{MAB}(\cdot, \cdot)$ computes attention between two sets. The time-complexity of an ISAB is $O(nm)$, scales linear with input set sizes.

## 3 METHODS

In this section, we present a novel extension of NPF called MPNPs. The main idea is to elicit joint predictive distributions that are constructed with equivariant neural networks instead of assuming priors for $\theta$, and let the corresponding martingale posterior describe the functional uncertainty in the NPs. We describe how we construct a MPNP in Section 3.1 and train it in Section 3.2.

## 3.1 MARTINGALE POSTERIOR NEURAL PROCESSES

Recall that the functional uncertainty in a NP is encoded in a parameter $\theta$. Rather than learning an approximate posterior $q(\theta|Z_c)$, we introduce a joint predictive $p(Z'|Z_c; \phi_{\text{pred}})$ generating a *pseudo context set* $Z' = \{z_i'\}_{i=1}^{N-|c|}$ of size $(N - |c|) \geq 1$. Having generated a pseudo context, we combine with the existing context $Z_c$, and construct the empirical density as

$$g_N(z) = \frac{1}{N} \left( \sum_{i \in c} \delta_{z_i}(z) + \sum_{i=1}^{N-|c|} \delta_{z_i'}(z) \right). \tag{10}$$

Given $g_N$, the estimate of the function parameter $\theta$ is then recovered as

$$\theta(g_N) := \arg\min_\theta \int \ell(z, \theta) g_N(dz), \tag{11}$$

where in our case we simply choose $\ell(z, \theta) := -\log \mathcal{N}(y|\mu_\theta(x), \sigma_\theta^2(x) I_{d_{\text{out}}})$. The uncertainty in $\theta(g_N)$ is thus induced by the uncertainty in the generated pseudo context $Z'$.

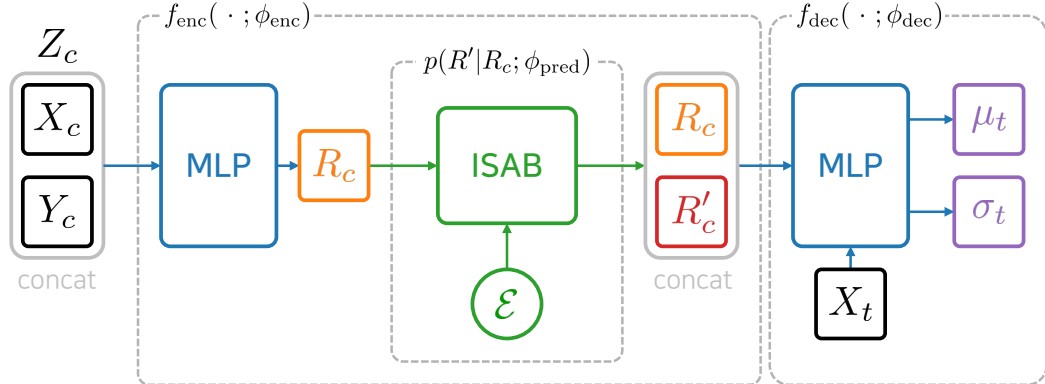

**Figure 1:** Concept figure of our feature generating model applied to CNP (Garnelo et al., 2018a). We first convert given context dataset $Z_c$ to the representation $R_c$ using Multi-Layer Perceptron (MLP) layers. Next we sample $\epsilon$ from a simple distribution (e.g. Gaussian). Then we generate the pseudo context representation $R'_c$ using generator as one layer ISAB (Lee et al., 2019) in our experiment.

**Amortization**  The procedure of recovering $\theta$ via Eq. 11 would originally require iterative optimization process except for simple cases. Fortunately, in our case, we can *amortize* this procedure, thanks to the mechanism of CNPs amortizing the inference procedure of estimating $\theta$ from the context. Given $Z_c$, a CNP learns an encoder producing $\theta$ that is trained to maximize the expected likelihood. That is,

$$\tilde{\theta}(Z_c) = f_{\text{enc}}(Z_c; \phi_{\text{enc}}), \quad \tilde{\theta}(Z_c) \approx \arg\min_{\theta} \int \ell(z, \theta) g_c(dz), \tag{12}$$

where $g_c$ is the empirical density of $Z_c$. Hence, given $Z'$ and $Z_c$, we can just input $Z' \cup Z_c$ into $f_{\text{enc}}$ and use the output $\tilde{\theta}(Z' \cup Z_c)$ as a proxy for $\theta(g_N)$. Compared to exactly computing $\theta(g_N)$, obtaining $\tilde{\theta}(Z' \cup Z_c)$ requires a single forward pass through $f_{\text{enc}}$, which scales much better with $N$. Moreover, computation for multiple $Z'$ required for bagging can easily be parallelized.

**Specifying the joint predictives**  We construct the joint predictives $p(Z'|Z_c; \phi_{\text{pred}})$ with a neural network. Other than the requirement of $Z'$ being exchangeable (and thus c.i.d.), we give no inductive bias to $p(Z'|Z_c; \phi_{\text{pred}})$ and let the model learn $\phi_{\text{pred}}$ from the data. We thus use the exchangeable generative model described in Section 2.4. Specifically, to generate $Z'$, we first generate $\mathcal{E} = \{\varepsilon_i\}_{i=1}^{n}$ from some distribution (usually chosen to be a unit Gaussian $\mathcal{N}(0, I_d)$), and pass them through an equivariant ISAB block to form $Z'$. To model the conditioning on $Z_c$, we set the inducing point in the ISAB as a transform of $Z_c$. That is, with an arbitrary feed-forward neural network $h$,

$$\text{ISAB}(\mathcal{E}) = \text{MAB}(\mathcal{E}, H), \quad H = \text{MAB}(h(Z_c), \mathcal{E}), \tag{13}$$

where $h(Z_c) = \{h(z_i)\}_{i \in c}$. The resulting model is an implicit generative model (Mohamed and Lakshminarayanan, 2016) in a sense that we can draw samples from it but cannot evaluate likelihoods.

**Generating Representations**  When $z$ is low-dimensional, it would be moderately easy to learn the joint predictives, but in practice, we often encounter problems with high-dimensional $z$, for instance when the input $x$ is a high-resolution image. For such cases, directly generating $z$ may be harder than the original problem, severely deteriorating the overall learning procedure of MPNP. Instead, we propose to generate the *encoded representations* of $z$. The encoders of the most of the NPFs first encode an input $z_i$ into a representation $r_i$. For the remaining of the forward pass, we only need $r_i$s instead of the original input $z$. Hence we can build a joint predictives $p(R'|R_c; \phi_{\text{pred}})$ generating $R' = \{r'_i\}_{i=1}^{N-|c|}$ conditioned on $R_c = \{r_i\}_{i \in c}$ as for generating $Z'$ from $Z_c$. In the experiments, we compare these two versions of MPNPs (generating $Z'$ and generating $R'$), and found that the one generating $R'$ works much better both in terms of data efficiency in training and predictive performances, even when the dimension of $z$ is not particularly large. See Fig. 1 for our method applying to CNP model (Garnelo et al., 2018a).

## 3.2 TRAINING

With the generator $p(Z'|Z_c; \phi_{\text{pred}})$, the marginal likelihood for a task $\tau = (Z, c)$ is computed as

$$\log p(Y|X, Z_c) = \log \int \exp \left( - \sum_{i \in [n]} \ell(z_i, \tilde{\theta}(Z_c \cup Z')) \right) p(Z'|Z_c; \phi_{\text{pred}}) \mathrm{d}Z'. \tag{14}$$

Note that $p(Z'|Z_c; \phi_{\text{pred}})$ is c.i.d., so there exists a corresponding martingale posterior $\pi_N$ such that

$$\log p(Y|X, Z_c) = \log \int \exp \left( - \sum_{i \in [n]} \ell(z_i, \theta) \right) \pi_N(\theta|Z_c) \mathrm{d}\theta. \tag{15}$$

We approximate the marginal likelihood via a consistent estimator,

$$\log p(Y|X, Z_c) \approx \log \left[ \frac{1}{K} \sum_{k=1}^{K} \exp \left( - \sum_{i \in [n]} \ell(z_i, \tilde{\theta}(Z_c \cup Z'^{(k)})) \right) \right] := -\mathcal{L}_{\text{marg}}(\tau, \phi), \tag{16}$$

where $Z'^{(1)}, \dots, Z'^{(K)} \overset{\text{i.i.d.}}{\sim} p(Z'|Z_c; \phi_{\text{pred}})$. This objective would be suffice if we are given sufficiently good $\tilde{\theta}(Z_c \cup Z'^{(k)})$, but we have to also train the encoder to properly amortize the parameter construction process Eq. 11. For this, we use only the given context data to optimize

$$\log p_{\text{CNP}}(Y|X, Z_c) = - \sum_{i \in [n]} \ell(z_i, \tilde{\theta}(Z_c)) := -\mathcal{L}_{\text{amort}}(\tau, \phi) \tag{17}$$

that is, we train the parameters $(\phi_{\text{enc}}, \phi_{\text{dec}})$ using CNP objective. Furthermore, we found that if we just maximize Eq. 16 and Eq. 17, the model can cheat by ignoring the generated pseudo contexts and use only the original context to build function estimates. To prevent this, we further maximize the similar CNP objectives for each generated pseudo context to encourage the model to actually make use of the generated contexts.

$$\frac{1}{K} \sum_{k=1}^{K} \log p_{\text{CNP}}(Y|X, Z'^{(k)}) = - \frac{1}{K} \sum_{i \in [n]} \ell(z_i, \tilde{\theta}(Z'^{(k)})) := -\mathcal{L}_{\text{pseudo}}(\tau, \phi) \tag{18}$$

Combining these, the loss function for the MPNP is then

$$\mathbb{E}_\tau[\mathcal{L}(\tau, \phi)] = \mathbb{E}_\tau[\mathcal{L}_{\text{marg}}(\tau, \phi) + \mathcal{L}_{\text{amort}}(\tau, \phi) + \mathcal{L}_{\text{pseudo}}(\tau, \phi)]. \tag{19}$$

## 4 RELATED WORKS

CNP (Garnelo et al., 2018a) is the first NPF model which consists of simple MLP layers as its encoder and decoder. NP (Garnelo et al., 2018b) also uses MLP layers as its encoder and decoder but introduces a global latent variable to model a functional uncertainty. Conditional Attentive Neural Process (CANP) (Kim et al., 2018) and Attentive Neural Process (ANP) (Kim et al., 2018) are the models which apply attention modules as their encoder block in order to well summarize context information relevant to target points. Louizos et al. (2019) proposed NPs model which employs local latent variables instead of a global latent variable by applying a graph neural network. By applying convolution layers as their encoder, Gordon et al. (2020) and Foong et al. (2020) introduced a translation equivariant CNPs and NPs model, respectively. In addition to these works, Bootstrapping Neural Process (BNP) (Lee et al., 2020) suggests modeling functional uncertainty with the bootstrap (Efron, 1992) method instead of using a single global latent variable.

## 5 EXPERIMENTS

We provide extensive experimental results to show how MPNP and Martingale Posterior Attentive Neural Process (MPANP) effectively increase performance upon the following baselines: CNP, NP, BNP, CANP, ANP, and Bootstrapping Attentive Neural Process (BANP). All models except deterministic models (i.e., CNP and CANP) use the same number of samples; $K = 5$ for the image completion task and $K = 10$ for the others. Refer to Appendices A and C for more detailed experimental setup including model architectures, dataset and evaluation metrics.

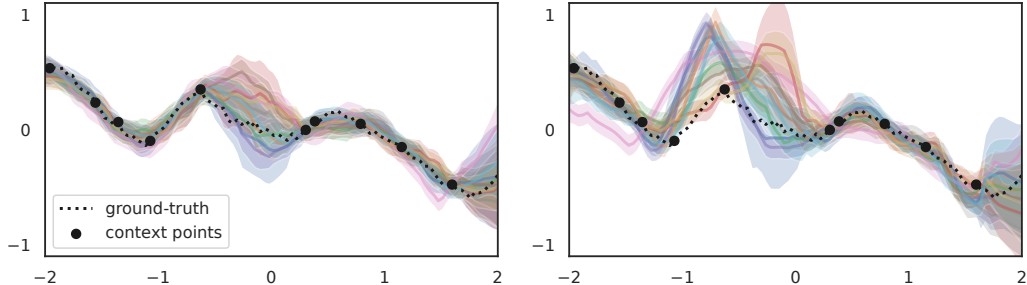

**Figure 2:** Posterior samples of MPANP for 1D regression task with RBF kernel. The black dashed line is the true function sampled from GP with RBF kernel, and the black dots are context points. We visualized decoded mean and standard deviation with colored lines and areas. (Left) MPANP posterior predictions using the combined features of the original contexts and the generated pseudo contexts. (Right) Predictions using only the generated pseudo contexts without the original contexts. The pseudo contexts are decoded into reasonable functions, especially with high uncertainty for the region without context points.

**Table 1:** Test results for 1D regression tasks on RBF, Matern, Periodic, and $t$-noise. 'Context' and 'Target' respectively denote context and target log-likelihood values. All values are averaged over four seeds. See Table 4 for the task log-likelihood values.

| Model | RBF | | Matern | | Periodic | | $t$-noise | |
|---|---|---|---|---|---|---|---|---|
| | Context | Target | Context | Target | Context | Target | Context | Target |
| CNP | $1.096_{\pm0.023}$ | $0.515_{\pm0.018}$ | $1.031_{\pm0.010}$ | $0.347_{\pm0.006}$ | $-0.120_{\pm0.020}$ | $-0.729_{\pm0.004}$ | $0.032_{\pm0.014}$ | $-0.816_{\pm0.032}$ |
| NP | $1.022_{\pm0.005}$ | $0.498_{\pm0.003}$ | $0.948_{\pm0.006}$ | $0.337_{\pm0.005}$ | $-0.267_{\pm0.024}$ | $\mathbf{-0.668}_{\pm0.006}$ | $\mathbf{0.201}_{\pm0.025}$ | $-0.333_{\pm0.078}$ |
| BNP | $1.112_{\pm0.003}$ | $0.588_{\pm0.004}$ | $1.057_{\pm0.009}$ | $0.418_{\pm0.006}$ | $-0.106_{\pm0.017}$ | $-0.705_{\pm0.001}$ | $-0.009_{\pm0.032}$ | $-0.619_{\pm0.191}$ |
| **MPNP (ours)** | $\mathbf{1.189}_{\pm0.005}$ | $\mathbf{0.675}_{\pm0.003}$ | $\mathbf{1.123}_{\pm0.005}$ | $\mathbf{0.481}_{\pm0.007}$ | $\mathbf{0.205}_{\pm0.020}$ | $\mathbf{-0.668}_{\pm0.008}$ | $0.145_{\pm0.017}$ | $\mathbf{-0.329}_{\pm0.025}$ |
| CANP | $1.304_{\pm0.027}$ | $0.847_{\pm0.005}$ | $1.264_{\pm0.041}$ | $0.662_{\pm0.013}$ | $0.527_{\pm0.106}$ | $-0.592_{\pm0.002}$ | $0.410_{\pm0.155}$ | $-0.577_{\pm0.022}$ |
| ANP | $\mathbf{1.380}_{\pm0.000}$ | $0.850_{\pm0.007}$ | $\mathbf{1.380}_{\pm0.000}$ | $0.663_{\pm0.004}$ | $0.583_{\pm0.011}$ | $-1.019_{\pm0.023}$ | $0.836_{\pm0.071}$ | $-0.415_{\pm0.131}$ |
| BANP | $\mathbf{1.380}_{\pm0.000}$ | $0.846_{\pm0.001}$ | $\mathbf{1.380}_{\pm0.000}$ | $0.662_{\pm0.005}$ | $\mathbf{1.354}_{\pm0.006}$ | $-0.496_{\pm0.005}$ | $0.646_{\pm0.042}$ | $-0.425_{\pm0.050}$ |
| **MPANP (ours)** | $1.379_{\pm0.000}$ | $\mathbf{0.881}_{\pm0.003}$ | $\mathbf{1.380}_{\pm0.000}$ | $\mathbf{0.692}_{\pm0.003}$ | $1.348_{\pm0.005}$ | $\mathbf{-0.494}_{\pm0.007}$ | $\mathbf{0.842}_{\pm0.062}$ | $\mathbf{-0.332}_{\pm0.026}$ |

## 5.1 1D REGRESSION

In this section, we conducted 1D regression experiments following Kim et al. (2018) and Lee et al. (2020). In this experiments, the dataset curves are generated from GP with 4 different settings: i) RBF kernels, ii) Matérn 5/2 kernels, iii) Periodic kernels, and iv) RBF kernels with Student's $t$ noise.

**Infinite Training Dataset** Previous works (Garnelo et al., 2018b; Kim et al., 2018; Le et al., 2018) assumed that there exists a GP curve generator that can provide virtually infinite amount of tasks for training. We first follow this setup, training all models for 100,000 steps where a new task is generated from each training step. We compare the models by picking checkpoints achieving the lowest validation loss. Table 1 clearly shows that our model outperforms the other models in most cases. This results show that our model well captures the functional uncertainty compared to the other methods. In Appendix B, we also report the comparison with the baselines with increased number of parameters to match the additional number of parameters introduced for the generator in our model, where ours still significantly outperforms the baselines.

**Finite Training Dataset** We also compare the models on more realistic setting assuming a finite amount of training tasks. Specifically, we first configured the finite training dataset consisting of $\{51200, 102400, 256000\}$ examples at the start of the training, instead of generating new tasks for each training step. We then trained all models with the same 100,000 training iterations in order to train the models with the same training budget as in the infinite training dataset situation. Fig. 3 clearly shows that our model consistently outperforms other models in terms of the target log-likelihood even when the training dataset is finite. This indicates that MPNPs effectively learn a predictive distribution of unseen dataset from a given dataset with small number of tasks. Refer to Appendix B for more detailed results.

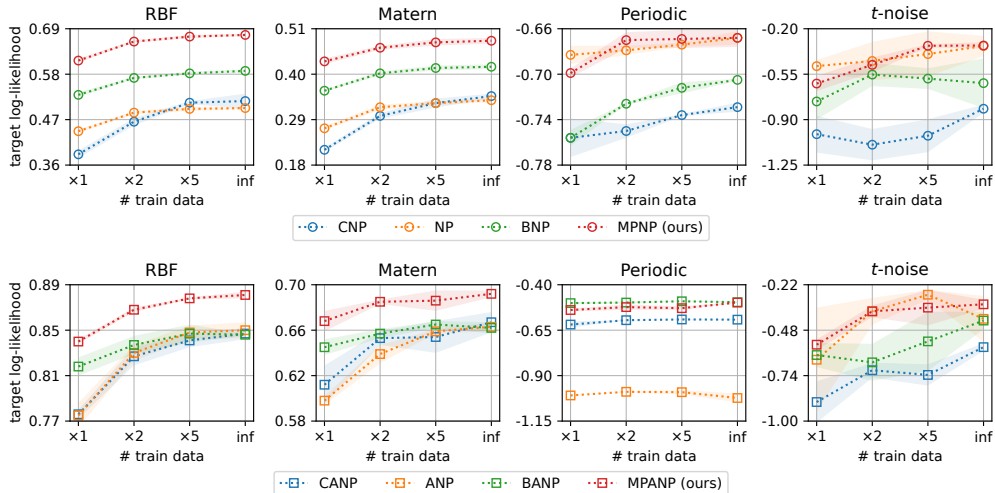

**Figure 3:** Test target log-likelihood values with varying the number of train data for 1D regression tasks on RBF, Matern, Periodic, and $t$-noise. Here, x-axis denotes how many examples are used for training, i.e., $\times 1$, $\times 2$, and $\times 5$ respectively denote 51200, 102400, and 256000 train examples.

**Table 2:** Test results for image completion tasks on MNIST, SVHN, and CelebA. 'Context' and 'Target' respectively denote context and target log-likelihood values, and 'Task' denotes the task log-likelihood. All values are averaged over four seeds.

| Model | MNIST | | | SVHN | | | CelebA | | |
|---|---|---|---|---|---|---|---|---|---|
| | Context | Target | Task | Context | Target | Task | Context | Target | Task |
| CNP | $0.878_{\pm 0.016}$ | $0.690_{\pm 0.010}$ | $0.706_{\pm 0.011}$ | $3.009_{\pm 0.069}$ | $2.785_{\pm 0.053}$ | $2.796_{\pm 0.054}$ | $2.692_{\pm 0.018}$ | $2.099_{\pm 0.011}$ | $2.134_{\pm 0.012}$ |
| NP | $0.797_{\pm 0.004}$ | $0.707_{\pm 0.004}$ | $0.714_{\pm 0.003}$ | $3.045_{\pm 0.021}$ | $2.841_{\pm 0.019}$ | $2.851_{\pm 0.019}$ | $2.721_{\pm 0.017}$ | $2.216_{\pm 0.013}$ | $2.246_{\pm 0.013}$ |
| BNP | $0.859_{\pm 0.050}$ | $0.742_{\pm 0.026}$ | $0.752_{\pm 0.029}$ | $3.169_{\pm 0.028}$ | $2.946_{\pm 0.023}$ | $2.957_{\pm 0.023}$ | $2.897_{\pm 0.011}$ | $2.329_{\pm 0.010}$ | $2.394_{\pm 0.010}$ |
| **MPNP (ours)** | $\mathbf{0.861}_{\pm 0.010}$ | $\mathbf{0.747}_{\pm 0.005}$ | $\mathbf{0.757}_{\pm 0.005}$ | $\mathbf{3.220}_{\pm 0.017}$ | $\mathbf{2.980}_{\pm 0.016}$ | $\mathbf{2.992}_{\pm 0.016}$ | $\mathbf{2.997}_{\pm 0.010}$ | $\mathbf{2.369}_{\pm 0.006}$ | $\mathbf{2.407}_{\pm 0.006}$ |
| CANP | $0.871_{\pm 0.020}$ | $0.688_{\pm 0.012}$ | $0.685_{\pm 0.013}$ | $3.079_{\pm 0.052}$ | $3.386_{\pm 0.020}$ | $3.335_{\pm 0.023}$ | $2.695_{\pm 0.033}$ | $2.674_{\pm 0.011}$ | $2.642_{\pm 0.011}$ |
| ANP | $1.186_{\pm 0.050}$ | $0.744_{\pm 0.008}$ | $0.793_{\pm 0.009}$ | $3.996_{\pm 0.064}$ | $3.365_{\pm 0.053}$ | $3.405_{\pm 0.053}$ | $4.086_{\pm 0.024}$ | $2.724_{\pm 0.029}$ | $2.833_{\pm 0.026}$ |
| BANP | $1.329_{\pm 0.021}$ | $0.752_{\pm 0.018}$ | $0.819_{\pm 0.018}$ | $4.019_{\pm 0.017}$ | $3.437_{\pm 0.026}$ | $3.476_{\pm 0.024}$ | $4.126_{\pm 0.003}$ | $2.764_{\pm 0.020}$ | $2.871_{\pm 0.018}$ |
| **MPANP (ours)** | $\mathbf{1.361}_{\pm 0.008}$ | $\mathbf{0.798}_{\pm 0.003}$ | $\mathbf{0.862}_{\pm 0.003}$ | $\mathbf{4.117}_{\pm 0.003}$ | $\mathbf{3.502}_{\pm 0.026}$ | $\mathbf{3.544}_{\pm 0.024}$ | $\mathbf{4.136}_{\pm 0.001}$ | $\mathbf{2.833}_{\pm 0.010}$ | $\mathbf{2.934}_{\pm 0.009}$ |

## 5.2 IMAGE COMPLETION

Next we conducted 2D image completion tasks for three different datasets, i.e., MNIST, SVHN, and CelebA. For training, we uniformly sample the number of context pixels $|c| \in \{3, ..., 197\}$ and the number of target pixels $|t| \in \{3, ..., 200 - |c|\}$ from an image. For evaluation, we uniformly sample the number of context pixels $|c| \in \{3, ..., 197\}$ and set all the remaining pixels as the targets. Table 2 clearly demonstrates that our model outperforms the baselines over all three datasets, demonstrating the effectiveness of our method for high-dimensional image data. See Appendix B for the visualizations of completed images along with the uncertainties in terms of predictive variances, and Appendix C for the detailed training setup.

## 5.3 BAYESIAN OPTIMIZATION

Using pre-trained models with RBF kernels in Section 5.1 Infinite Training Dataset experiments, we conducted Bayesian optimization (Brochu et al., 2010) for two benchmark functions (Gramacy and Lee, 2012; Forrester et al., 2008). As a performance measurement, we use best simple regret, which measures the difference between the current best value and the global optimum value. Fig. 4 depicts the normalized regret and the cumulative normalized regret averaged over 100 trials of the Gramacy and Lee (2012) function. Here, we also consider a GP variant with RBF kernel, tuned by pre-training (Wang et al., 2022). It clearly demonstrates that our model shows the best performance among NPs for both the normalized regret and the cumulative normalized regret. Appendix B.4 provides the results for the Forrester et al. (2008) function and Appendix C.4 provides detailed experimental setups.

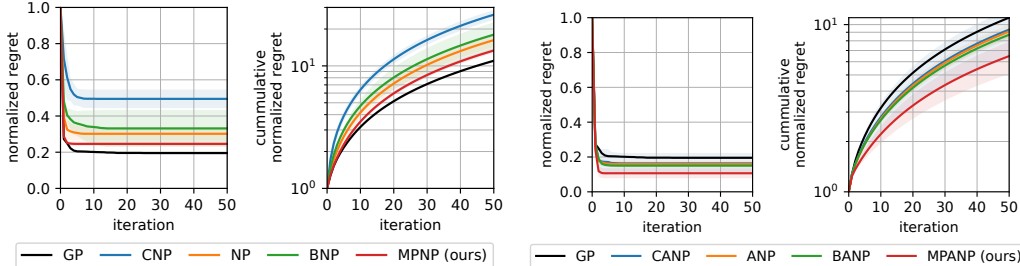

**Figure 4:** Results for Bayesian optimization on Gramacy and Lee (2012) function; we measured normalized simple regret and its cumulative value for a iteration. All models are pre-trained on 1D regression task generated with RBF kernel (cf. Section 5.1) and evaluated on the benchmark function for Bayesian optimization.

**Table 3:** Test results for predator-prey population regression tasks on Lotka-Volterra simulated data and real data. 'Context' and 'Target' respectively denote context and target log-likelihood values, and 'Task' denotes the task log-likelihood. All values are averaged over four seeds.

| | Simulated data | | | Real data | | |
|---|---|---|---|---|---|---|
| Model | Context | Target | Task | Context | Target | Task |
| CNP | $0.327_{\pm0.036}$ | $0.035_{\pm0.029}$ | $0.181_{\pm0.032}$ | $-2.686_{\pm0.024}$ | $-3.201_{\pm0.042}$ | $-3.000_{\pm0.034}$ |
| NP | $0.112_{\pm0.063}$ | $-0.115_{\pm0.057}$ | $0.000_{\pm0.060}$ | $-2.770_{\pm0.028}$ | $-3.144_{\pm0.031}$ | $-2.993_{\pm0.029}$ |
| BNP | $0.550_{\pm0.057}$ | $0.274_{\pm0.042}$ | $0.417_{\pm0.050}$ | $\mathbf{-2.614}_{\pm0.050}$ | $\mathbf{-3.052}_{\pm0.022}$ | $\mathbf{-2.868}_{\pm0.024}$ |
| **MPNP (ours)** | $\mathbf{0.626}_{\pm0.041}$ | $\mathbf{0.375}_{\pm0.036}$ | $\mathbf{0.500}_{\pm0.038}$ | $-2.621_{\pm0.072}$ | $-3.092_{\pm0.054}$ | $-2.918_{\pm0.061}$ |
| CANP | $0.689_{\pm0.046}$ | $1.615_{\pm0.026}$ | $1.023_{\pm0.018}$ | $-4.743_{\pm1.119}$ | $-6.413_{\pm0.339}$ | $-5.801_{\pm0.733}$ |
| ANP | $2.607_{\pm0.015}$ | $1.830_{\pm0.020}$ | $2.234_{\pm0.018}$ | $1.887_{\pm0.078}$ | $-4.848_{\pm0.385}$ | $-1.615_{\pm0.188}$ |
| BANP | $\mathbf{2.654}_{\pm0.000}$ | $1.797_{\pm0.012}$ | $2.240_{\pm0.006}$ | $\mathbf{2.190}_{\pm0.062}$ | $\mathbf{-3.597}_{\pm0.279}$ | $\mathbf{-0.741}_{\pm0.160}$ |
| **MPANP (ours)** | $2.639_{\pm0.008}$ | $\mathbf{1.835}_{\pm0.004}$ | $\mathbf{2.254}_{\pm0.006}$ | $1.995_{\pm0.145}$ | $-5.073_{\pm0.680}$ | $-1.690_{\pm0.401}$ |

## 5.4 PREDATOR-PREY MODEL

Following Lee et al. (2020), we conducted the predator-prey population regression experiments. We first trained the models using the simulation datasets which are generated from a Lotka-Volterra model (Wilkinson, 2018) with the simulation settings followed by Lee et al. (2020). Then tested on the generated simulation test dataset and real-world dataset which is called Hudson's Bay hare-lynx data. As mentioned in Lee et al. (2020), the real-world dataset shows different tendency from generated simulation datasets, so we can treat this experiment as model-data mismatch experiments. In Table 3, we can see the MPNPs outperform the other baselines for the test simulation datasets but underperforms in the real-world dataset compare to other baselines. This shows that model-data mismatch is an open problem for the MPNPs.

## 6 CONCLUSION

In this paper, we proposed a novel extension of NPs by taking a new approach to model the functional uncertainty for NPs. The proposed model MPNP utilizes the martingale posterior distribution (Fong et al., 2021), where the functional uncertainty is driven from the uncertainty of future data generated from the joint predictive. We present a simple architecture satisfying the theoretical requirements of the martingale posterior, and propose a training scheme to properly train it. We empirically validate MPNPs on various tasks, where our method consistently outperforms the baselines.

**Limitation** As we presented in the **Predator-Prey Model** experiments in Section 5.4, our method did not significantly outperform baselines under model-data mismatch. This was also highlighted in Fong et al. (2021): model-data mismatch under the martingale posterior framework remains an open problem. Our method with direct input generation also performed poorly, as we found it difficult to prevent models from generating meaningless inputs that are ignored by the decoders. We present more details on unsuccessful attempts for direct input generation in Appendix D.

**Societal Impacts**   Our work is unlikely to bring any negative societal impacts. Modeling functional uncertainty may be related to the discussion of safe AI within the community.

**Reproducibility Statement**   We argued our experimental details in Appendix C which contains used libraries and hardwares. We presented all the dataset description in Appendix C. We describes the model architecture details in Appendix A.

## ACKNOWLEDGEMENTS

This work was partly supported by Institute of Information & communications Technology Planning & Evaluation (IITP) grant funded by the Korea government(MSIT) (No.2019-0-00075, Artificial Intelligence Graduate School Program(KAIST)), Institute of Information & communications Technology Planning & Evaluation (IITP) grant funded by the Korea government(MSIT) (No.2022-0-00713), and Institute of Information & communications Technology Planning & Evaluation (IITP) grant funded by the Korea government(MSIT) (No.2021-0-02068, Artificial Intelligence Innovation Hub).

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

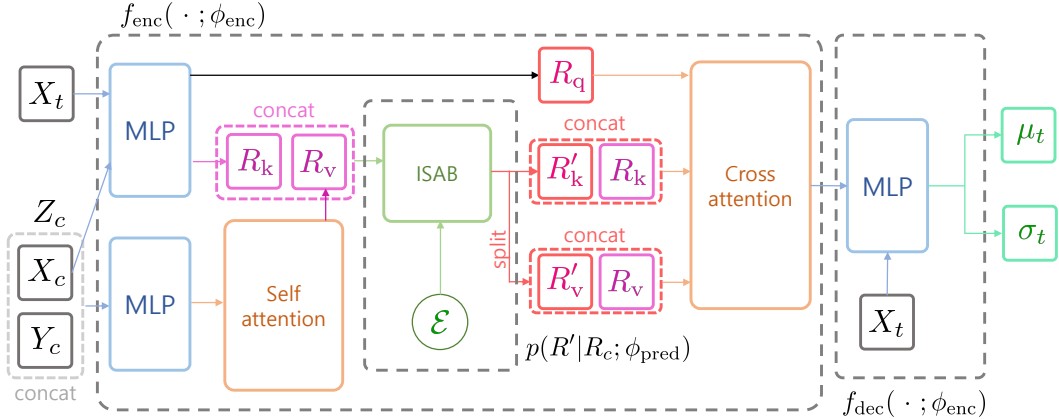

**Figure 5:** Concept figure of our feature generating model applied to CANP (Kim et al., 2018). Here we sample $\epsilon$ from a simple distribution (e.g. Gaussian). We generate key feature $R'_k$ and value feature $R'_v$ for cross attention layer which are corresponding to pseudo context data. We use generator as one layer ISAB (Lee et al., 2019) in our experiment.

## A MODEL ARCHITECTURES

In this section, we summarize the model architectures which we used in experiments. Here, we only present simplified structures for each model. To see exact computation procedures for BNPs, please refer to Lee et al. (2020). Fig. 5 shows our method applying to CANP model (Kim et al., 2018).

### A.1 MODULES

**Linear Layer** $\text{Lin}(d_{\text{in}}, d_{\text{out}})$ denotes the linear transformation of the input with dimension $d_{\text{in}}$ into the output with dimension $d_{\text{out}}$.

**Multi-Layer Perceptron** $\text{MLP}(n_l, d_{\text{in}}, d_{\text{hid}}, d_{\text{out}})$ denotes a multi-layer perceptron with the structure:

$$\text{MLP}(n_l, d_{\text{in}}, d_{\text{hid}}, d_{\text{out}}) = \text{Lin}(d_{\text{hid}}, d_{\text{out}}) \circ (\text{ReLU} \circ \text{Lin}(d_{\text{hid}}, d_{\text{hid}}))^{n_l - 2} \circ \text{ReLU} \circ \text{Lin}(d_{\text{in}}, d_{\text{hid}}),$$

where ReLU denotes the element-wise Rectified Linear Unit (ReLU) activation function.

**Multi-Head Attention** $\text{MHA}(n_{\text{head}}, d_{\text{out}})(Q, K, V)$ denotes a multi-head attention (Vaswani et al., 2017) with $n_{\text{head}}$ heads which takes input as $(Q, K, V)$ and outputs the feature with dimension $d_{\text{out}}$. The actual computation of $\text{MHA}(n_{\text{head}}, d_{\text{out}})(Q, K, V)$ can be written as follows:

$$(Q'_i)_{i=1}^{n_{\text{head}}} = \text{split}(\text{Lin}(d_q, d_{\text{out}})(Q), n_{\text{head}})$$
$$(K'_i)_{i=1}^{n_{\text{head}}} = \text{split}(\text{Lin}(d_k, d_{\text{out}})(K), n_{\text{head}})$$
$$(V'_i)_{i=1}^{n_{\text{head}}} = \text{split}(\text{Lin}(d_v, d_{\text{out}})(V), n_{\text{head}})$$
$$H = \text{concat}([\text{softmax}(Q'_i K'^{\top}_i / \sqrt{d_{\text{out}}}) V'_i]_{i=1}^{n_{\text{head}}})$$
$$O = \text{LN}(Q' + H)$$
$$\text{MHA}(n_{\text{head}}, d_{\text{out}})(Q, K, V) = \text{LN}(O + \text{ReLU}(\text{Lin}(d_{\text{out}}, d_{\text{out}})(O)))$$

where $(d_q, d_k, d_v)$ denotes the dimension of $Q, K, V$ respectively, split and concat are the splitting and concatenating $A$ in the feature dimension respectively, and LN denotes the layer normalization (Ba et al., 2016).

**Self-Attention** $\text{SA}(n_{\text{head}}, d_{\text{out}})$ denotes a self-attention module which is simply computed as $\text{SA}(n_{\text{head}}, d_{\text{out}})(X) = \text{MHA}(n_{\text{head}}, d_{\text{out}})(X, X, X)$.

**Multi-head Attention Block** $\text{MAB}(n_{\text{head}}, d_{\text{out}})$ denotes a multi-head attention block module (Lee et al., 2019) which is simply computed as $\text{MAB}(n_{\text{head}}, d_{\text{out}})(X, Y) = \text{MHA}(n_{\text{head}}, d_{\text{out}})(X, Y, Y)$.

**Induced Set Attention Block** $\text{ISAB}(n_{\text{head}}, d_{\text{out}})$ denotes a induced set attention block (Lee et al., 2019) which constructed with two stacked MAB layers. The actual computation of $\text{ISAB}(n_{\text{head}}, d_{\text{out}})(X, Y)$ can be written as follows:

$$H = \text{MAB}(n_{\text{head}}, d_{\text{out}})(Y, X)$$
$$\text{ISAB}(n_{\text{head}}, d_{\text{out}})(X, Y) = \text{MAB}(n_{\text{head}}, d_{\text{out}})(X, H).$$

A.2 CNP, NP, BNP, NEURAL BOOTSTRAPPING NEURAL PROCESS (NEUBNP) AND MPNP

**Encoder** The models only with a deterministic encoder (CNP, BNP, MPNP) use the following structure:

$$r_c = \frac{1}{|c|} \sum_{i \in c} \text{MLP}(n_l = 5, d_{\text{in}} = d_z, d_{\text{hid}} = 128, d_{\text{out}} = 128)(z_i),$$
$$f_{\text{enc}}(Z_c) = r_c.$$

For the MPNP, $f_{\text{enc}}(Z_c)$ changes into $\text{concat}([r_c, r'_c])$ where $r'_c$ is the feature of the pseudo context data generated from generator in paragraph **Generator**. The model also with a latent encoder (NP) uses:

$$r_c = \frac{1}{|c|} \sum_{i \in c} \text{MLP}(n_l = 5, d_{\text{in}} = d_z, d_{\text{hid}} = 128, d_{\text{out}} = 128)(z_i),$$
$$(m_c, \log s_c) = \frac{1}{|c|} \sum_{i \in c} \text{MLP}(n_l = 2, d_{\text{in}} = d_z, d_{\text{hid}} = 128, d_{\text{out}} = 128 \times 2)(z_i),$$
$$s_c = 0.1 + 0.9 \cdot \text{softplus}(\log s_c),$$
$$h_c = \mathcal{N}(m_c, s_c^2 I_h),$$
$$f_{\text{enc}}(Z_c) = [r_c; h_c],$$

where $d_z = d_x + d_y$ denotes the data dimension. Data dimensions vary through tasks, $d_x = 1, d_y = 1$ for 1D regression tasks, $d_x = 2, d_y = 1$ for MNIST image completion task, $d_x = 2, d_y = 3$ for SVHN and CelebA image completion tasks, and $d_x = 1, d_y = 2$ for Lotka Volterra task.

**Adaptation Layer** BNP uses additional adaptation layer to combine bootstrapped representation and the base representation. This can be done with a simple linear layer

$$\tilde{r}_c = \text{Lin}(d_{\text{hid}} = 128, d_{\text{hid}} = d_x + 128)(\tilde{r}_c^{(pre)}). \tag{20}$$

**Decoder** All models use a single MLP as a decoder. The models except NP uses the following structure:

$$(\mu, \log \sigma) = \text{MLP}(n_l = 3, d_{\text{in}} = d_x + 128, d_{\text{hid}} = 128, d_{\text{out}} = 2)(\text{concat}([x, r_c]))$$
$$\sigma = 0.1 + 0.9 \cdot \text{softplus}(\log \sigma),$$
$$f_{\text{dec}}(x, r_c) = (\mu, \sigma),$$

and NP uses:

$$(\mu, \log \sigma) = \text{MLP}(n_l = 3, d_{\text{in}} = d_x + 128 \times 2, d_{\text{hid}} = 128, d_{\text{out}} = 2)(\text{concat}([x, r_c, h_c]))$$
$$\sigma = 0.1 + 0.9 \cdot \text{softplus}(\log \sigma),$$
$$f_{\text{dec}}(x, r_c, h_c) = (\mu, \sigma).$$

**Generator** MPNP use a single ISAB module as a generator. The ISAB uses the following structure:

$$\epsilon = \text{concat}([\epsilon_i]_{i=1}^{n_{\text{gen}}})$$
$$r'_c = \text{ISAB}(n_{\text{head}} = 8, d_{\text{out}} = 128)(\epsilon, r_c)$$
$$f_{\text{gen}}(r_c) = r'_c$$

where $\epsilon_i$s are i.i.d. sampled from Gaussian distribution with dimension 128 and $n_{\text{gen}}$ denotes a number of pseudo context data.

### A.3 CANP, ANP, BANP AND MPANP

**Encoder** The models only with a deterministic encoder (CANP, BANP, Neural Bootstrapping Attentive Neural Process (NEUBANP) and MPANP) use the following structure:

$$r_q = \text{MLP}(n_l = 5, d_{\text{in}} = d_x, d_{\text{hid}} = 128, d_{\text{out}} = 128)(X),$$
$$r_k = \text{MLP} \qquad\qquad '' \qquad\qquad (X_c),$$
$$r_v^{(\text{pre})} = \text{MLP}(n_l = 5, d_{\text{in}} = d_z, d_{\text{hid}} = 128, d_{\text{out}} = 128)(X_c),$$
$$r_v = \text{SA}(n_{\text{head}} = 8, d_{\text{out}} = 128)(r_v^{(\text{pre})}),$$
$$r_c = \text{MHA}(n_{\text{head}} = 8, d_{\text{out}} = 128)(r_q, r_k, r_v),$$
$$f_{\text{enc}}(Z_c) = r_c.$$

For the MPANP, $f_{\text{enc}}(Z_c)$ changes into

$$r_c = \text{MHA}(n_{\text{head}} = 8, d_{\text{out}} = 128)(r_q, \text{concat}([r_k, r_k']), \text{concat}([r_v, r_v'])),$$
$$f_{\text{enc}} = r_c,$$

where $r_k'$ and $r_v'$ are the key and value features of the pseudo context data generated from generator in paragraph **Generator**.

ANP constructed as:

$$r_q = \text{MLP}(n_l = 5, d_{\text{in}} = d_x, d_{\text{hid}} = 128, d_{\text{out}} = 128)(X),$$
$$r_k = \text{MLP} \qquad\qquad '' \qquad\qquad (X_c),$$
$$r_v' = \text{MLP}(n_l = 5, d_{\text{in}} = d_z, d_{\text{hid}} = 128, d_{\text{out}} = 128)(X_c),$$
$$r_v = \text{SA}(n_{\text{head}} = 8, d_{\text{out}} = 128)(r_v'),$$
$$r_c = \text{MHA}(n_{\text{head}} = 8, d_{\text{out}} = 128)(r_q, r_k, r_v),$$
$$h_i' = \text{MLP}(n_l = 2, d_{\text{in}} = d_z, d_{\text{hid}} = 128, d_{\text{out}} = 128 \times 2)(z_i),$$
$$h_i = \text{SA}(n_{\text{head}} = 8, d_{\text{out}} = 128)(h_i'),$$
$$(m_c, \log s_c) = \frac{1}{|c|} \sum_{i \in c} h_i,$$
$$s_c = 0.1 + 0.9 \cdot \text{softplus}(\log s_c),$$
$$h_c = \mathcal{N}(m_c, s_c^2 I_h),$$
$$f_{\text{enc}}(Z_c) = [r_c; h_c].$$

Note that $r_q$ and $r_k$ are from the same MLP.

**Adaptation Layer** Like BNP, BANP also uses adaptation layer with same structure to combine bootstrapped representations.

**Decoder** All models use the same decoder structure as their non-attentive counterparts.

**Generator** MPANP use a single ISAB module as a generator. The ISAB uses the following structure:

$$\epsilon = \text{concat}([\epsilon_i]_{i=1}^{n_{\text{gen}}})$$
$$(r_k', r_v') = \text{ISAB}(n_{\text{head}} = 8, d_{\text{out}} = 256)(\epsilon, \text{concat}([r_k, r_v]))$$
$$f_{\text{gen}}(r_k, r_v) = (r_k', r_v')$$

where $\epsilon_i$s are i.i.d. sampled from Gaussian distribution with dimension 256 and $n_{\text{gen}}$ denotes a number of pseudo context data.

## B ADDITIONAL EXPERIMENTS

### B.1 1D REGRESSION

**Full results for Table 1** We provide the full test results for 1D regression tasks including context, target, and task log-likelihood values in Table 4.

**Table 4:** Test results for 1D regression tasks on RBF, Matern, Periodic, and $t$-noise. 'Context' and 'Target' respectively denote context and target log-likelihood values, and 'Task' denotes the task log-likelihood. All values are averaged over four seeds.

| Model | RBF | | | Matern | | | Periodic | | | $t$-noise | | |
|---|---|---|---|---|---|---|---|---|---|---|---|---|
| | Context | Target | Task | Context | Target | Task | Context | Target | Task | Context | Target | Task |
| CNP | $1.096_{\pm0.023}$ | $0.515_{\pm0.018}$ | $0.796_{\pm0.020}$ | $1.031_{\pm0.010}$ | $0.347_{\pm0.006}$ | $0.693_{\pm0.008}$ | $-0.120_{\pm0.020}$ | $-0.729_{\pm0.004}$ | $-0.363_{\pm0.012}$ | $0.032_{\pm0.014}$ | $-0.816_{\pm0.032}$ | $-0.260_{\pm0.012}$ |
| NP | $1.022_{\pm0.005}$ | $0.498_{\pm0.003}$ | $0.748_{\pm0.004}$ | $0.948_{\pm0.006}$ | $0.337_{\pm0.005}$ | $0.641_{\pm0.005}$ | $-0.267_{\pm0.024}$ | $\mathbf{-0.668}_{\pm0.006}$ | $-0.441_{\pm0.013}$ | $\mathbf{0.201}_{\pm0.025}$ | $-0.333_{\pm0.078}$ | $\mathbf{-0.038}_{\pm0.026}$ |
| BNP | $1.112_{\pm0.003}$ | $0.588_{\pm0.004}$ | $0.841_{\pm0.003}$ | $1.057_{\pm0.009}$ | $0.418_{\pm0.006}$ | $0.741_{\pm0.007}$ | $-0.106_{\pm0.017}$ | $-0.705_{\pm0.001}$ | $-0.347_{\pm0.010}$ | $-0.009_{\pm0.032}$ | $-0.619_{\pm0.191}$ | $-0.217_{\pm0.036}$ |
| **MPNP (ours)** | $\mathbf{1.189}_{\pm0.005}$ | $\mathbf{0.675}_{\pm0.003}$ | $\mathbf{0.911}_{\pm0.003}$ | $\mathbf{1.123}_{\pm0.005}$ | $\mathbf{0.481}_{\pm0.007}$ | $\mathbf{0.796}_{\pm0.005}$ | $\mathbf{0.205}_{\pm0.020}$ | $\mathbf{-0.668}_{\pm0.008}$ | $\mathbf{-0.171}_{\pm0.013}$ | $0.145_{\pm0.017}$ | $\mathbf{-0.329}_{\pm0.025}$ | $-0.061_{\pm0.012}$ |
| CANP | $1.304_{\pm0.027}$ | $0.847_{\pm0.005}$ | $1.036_{\pm0.020}$ | $1.264_{\pm0.041}$ | $0.662_{\pm0.013}$ | $0.937_{\pm0.031}$ | $0.527_{\pm0.106}$ | $-0.592_{\pm0.002}$ | $0.010_{\pm0.069}$ | $0.410_{\pm0.155}$ | $-0.577_{\pm0.022}$ | $-0.008_{\pm0.098}$ |
| ANP | $\mathbf{1.380}_{\pm0.000}$ | $0.850_{\pm0.007}$ | $1.090_{\pm0.003}$ | $\mathbf{1.380}_{\pm0.000}$ | $0.663_{\pm0.004}$ | $1.019_{\pm0.002}$ | $0.583_{\pm0.011}$ | $-1.019_{\pm0.023}$ | $0.090_{\pm0.004}$ | $\mathbf{0.836}_{\pm0.071}$ | $-0.415_{\pm0.131}$ | $0.374_{\pm0.034}$ |
| BANP | $\mathbf{1.380}_{\pm0.000}$ | $0.846_{\pm0.001}$ | $1.088_{\pm0.000}$ | $\mathbf{1.380}_{\pm0.000}$ | $0.662_{\pm0.005}$ | $1.018_{\pm0.002}$ | $\mathbf{1.354}_{\pm0.006}$ | $-0.496_{\pm0.005}$ | $\mathbf{0.634}_{\pm0.005}$ | $0.646_{\pm0.042}$ | $-0.425_{\pm0.050}$ | $0.270_{\pm0.033}$ |
| **MPANP (ours)** | $1.379_{\pm0.000}$ | $\mathbf{0.881}_{\pm0.003}$ | $\mathbf{1.102}_{\pm0.001}$ | $\mathbf{1.380}_{\pm0.000}$ | $\mathbf{0.692}_{\pm0.003}$ | $\mathbf{1.029}_{\pm0.001}$ | $1.348_{\pm0.005}$ | $\mathbf{-0.494}_{\pm0.007}$ | $0.630_{\pm0.005}$ | $\mathbf{0.842}_{\pm0.062}$ | $\mathbf{-0.332}_{\pm0.026}$ | $\mathbf{0.384}_{\pm0.041}$ |

**Table 5:** Further comparisons with baselines with increased number of parameters. 'Context' and 'Target' respectively denote context and target log-liklihood values, and 'Task' denotes the task log-likelihood. All values are averaged over four seeds.

| Model | # Params | RBF | | | Matern | | |
|---|---|---|---|---|---|---|---|
| | | Context | Target | Task | Context | Target | Task |
| CNP | 264 K | $1.096_{\pm0.008}$ | $0.517_{\pm0.007}$ | $0.797_{\pm0.007}$ | $1.017_{\pm0.021}$ | $0.340_{\pm0.012}$ | $0.681_{\pm0.017}$ |
| NP | 274 K | $1.026_{\pm0.004}$ | $0.501_{\pm0.003}$ | $0.752_{\pm0.003}$ | $0.948_{\pm0.005}$ | $0.334_{\pm0.002}$ | $0.640_{\pm0.003}$ |
| BNP | 261 K | $1.115_{\pm0.007}$ | $0.591_{\pm0.005}$ | $0.843_{\pm0.006}$ | $1.051_{\pm0.007}$ | $0.416_{\pm0.005}$ | $0.736_{\pm0.005}$ |
| **MPNP (ours)** | 266 K | $\mathbf{1.189}_{\pm0.005}$ | $\mathbf{0.675}_{\pm0.003}$ | $\mathbf{0.911}_{\pm0.003}$ | $\mathbf{1.123}_{\pm0.005}$ | $\mathbf{0.481}_{\pm0.007}$ | $\mathbf{0.796}_{\pm0.005}$ |
| CANP | 868 K | $1.305_{\pm0.007}$ | $0.844_{\pm0.006}$ | $1.035_{\pm0.005}$ | $1.278_{\pm0.013}$ | $0.663_{\pm0.006}$ | $0.947_{\pm0.008}$ |
| ANP | 877 K | $\mathbf{1.380}_{\pm0.000}$ | $0.858_{\pm0.002}$ | $1.093_{\pm0.001}$ | $\mathbf{1.380}_{\pm0.000}$ | $0.668_{\pm0.006}$ | $1.020_{\pm0.002}$ |
| BANP | 885 K | $1.379_{\pm0.001}$ | $0.839_{\pm0.015}$ | $1.085_{\pm0.007}$ | $1.376_{\pm0.005}$ | $0.652_{\pm0.032}$ | $1.012_{\pm0.014}$ |
| **MPANP (ours)** | 877 K | $1.379_{\pm0.000}$ | $\mathbf{0.881}_{\pm0.003}$ | $\mathbf{1.102}_{\pm0.001}$ | $\mathbf{1.380}_{\pm0.000}$ | $\mathbf{0.692}_{\pm0.003}$ | $\mathbf{1.029}_{\pm0.001}$ |

**Increasing the encoder size of baselines** Since the generator increases the size of the encoder in MPNPs, one can claim that the performance gain of MPNPs may come from the increased model size. To verify this, we increased the hidden dimensions of the encoder of baselines and compared them with ours. The results displayed in Table 5 further clarify that ours still outperforms the baselines even when the number of parameters gets in line.

## B.2 HIGH-D REGRESSION

We conducted additional experiments on the synthetic high-dimensional regression data (i.e., generating one-dimensional y from four-dimensional x with RBF kernel). Here we used the same model structures with the 1D regression task except for the input layer, and the same settings for the RBF kernel with 1D regression except for $l \sim \text{Unif}(0.5, 3.0)$. We fixed the base learning rate to $0.00015$ for all models throughout the high-dimensional regression experiments.

Table 6 clearly shows our MPNPs still outperform baselines for log-likelihood values we measured.

## B.3 IMAGE COMPLETION

**MNIST** We provide some completed MNIST images in Fig. 6. It shows that both MPNP and MPANP successfully fill up the remaining parts of the image for a given context and capture the uncertainties as predictive variances.

**CelebA** We also present five examples from the CelebA dataset in Fig. 7. It shows that MPANP provides perceptually reasonable predictions even for complex three-channel images.

## B.4 BAYESIAN OPTIMIZATION

We provide the results for Bayesian optimization on the Forrester et al. (2008) function in Fig. 8. Our MPNPs consistently outperform baselines as discussed in Section 5.3. We also present the visual results for Bayesian optimization in Figs. 9 and 10.

**Table 6:** Test results for 4D regression tasks on RBF. 'Context' and 'Target' respectively denote context and target log-likelihood values, and 'Task' denotes the task log-likelihood. All values are averaged over four seeds.

| Model | RBF | | |
|---|---|---|---|
| | Context | Target | Task |
| CNP | $0.572_{\pm 0.003}$ | $0.265_{\pm 0.002}$ | $0.410_{\pm 0.003}$ |
| NP | $0.568_{\pm 0.009}$ | $0.267_{\pm 0.004}$ | $0.407_{\pm 0.007}$ |
| BNP | $0.621_{\pm 0.015}$ | $0.323_{\pm 0.008}$ | $0.467_{\pm 0.013}$ |
| **MPNP (ours)** | $\mathbf{0.820}_{\pm 0.002}$ | $\mathbf{0.441}_{\pm 0.004}$ | $\mathbf{0.633}_{\pm 0.004}$ |
| CANP | $0.957_{\pm 0.005}$ | $0.585_{\pm 0.006}$ | $0.743_{\pm 0.005}$ |
| ANP | $1.357_{\pm 0.006}$ | $0.320_{\pm 0.014}$ | $0.890_{\pm 0.007}$ |
| BANP | $\mathbf{1.380}_{\pm 0.000}$ | $0.549_{\pm 0.006}$ | $1.013_{\pm 0.002}$ |
| **MPANP (ours)** | $1.379_{\pm 0.000}$ | $\mathbf{0.645}_{\pm 0.007}$ | $\mathbf{1.046}_{\pm 0.002}$ |

## C   EXPERIMENTAL DETAILS

We attached our code in supplementary material. Our codes used python libraries JAX (Bradbury et al., 2018), Flax (Heek et al., 2020) and Optax (Hessel et al., 2020). These python libraries are available under the Apache-2.0 license[1].

We conducted all experiments on a single NVIDIA GeForce RTX 3090 GPU, except for the image completion tasks presented in Section 5.2; we used 8 TPUv3 cores supported by TPU Research Cloud[2] for the 2D image completion task. For optimization, we used Adam (Kingma and Ba, 2015) optimizer with a cosine learning rate schedule. Unless specified, we selected the base learning rate from a grid of $\{5 \times 10^{-4.50}, 5 \times 10^{-4.25}, 5 \times 10^{-4.00}, 5 \times 10^{-3.75}, 5 \times 10^{-3.50}\}$ based on validation task log-likelihood.

### C.1   EVALUATION METRIC

Following Le et al. (2018), for CNP and CANP, which are deterministic models, we used the normalized predictive log-likelihood $\frac{1}{n}\sum_{i=1}^{n} \log p(y_i|x_i, Z_c)$. For other models, we used a approximation of the normalized predictive log-likelihood as:

$$\frac{1}{n}\sum_{i=1}^{n} \log p(y_i|x_i, Z_c) \approx \frac{1}{n}\sum_{i=1}^{n} \log \frac{1}{K}\sum_{k=1}^{K} p(y_i|x_i, \theta^{(k)}), \tag{21}$$

where $\theta^k$s are independent samples for $k \in [K]$.

### C.2   1D REGRESSION

To generate tasks $(Z, c)$, we first sample $x \overset{\text{i.i.d.}}{\sim} \text{Unif}(-2, 2)$ and generate $Y$ using each kernel. We use RBF kernel $k(x, x') = s^2 \cdot \exp\left(\frac{-||x - x'||^2}{2\ell^2}\right)$, Matern 5/2 kernel $k(x, x') = s^2 \cdot \left(1 + \frac{\sqrt{5}d}{\ell} + \frac{5d^2}{3\ell^2}\right)$, and periodic kernel $k(x, x') = s^2 \cdot \exp\left(\frac{-2\sin^2(\pi||x-x'||^2/p)}{\ell^2}\right)$ where all kernels use $s \sim \text{Unif}(0.1.1.0)$, $\ell \sim \text{Unif}(0.1.0.6)$, and $p \sim \text{Unif}(0.1.0.5)$. To generate t-noise dataset, we use Student-$t$ with degree of freedom 2.1 to sample noise $\epsilon \sim \gamma \cdot \mathcal{T}(2.1)$ where $\gamma \sim \text{Unif}(0, 0.15)$. Then we add the noise to the curves generated from RBF kernel. We draw index set $|c| \sim \text{Unif}(3, 50 - 3)$ and $n - |c| \sim \text{Unif}(3, 50 - |c|)$ to maintain $\max |Z| \leq 50$. We use a batch size of 256 for training.

### C.3   IMAGE COMPLETION

We use the following datasets for image completion experiments.

---

[1] https://www.apache.org/licenses/LICENSE-2.0
[2] https://sites.research.google/trc/about/

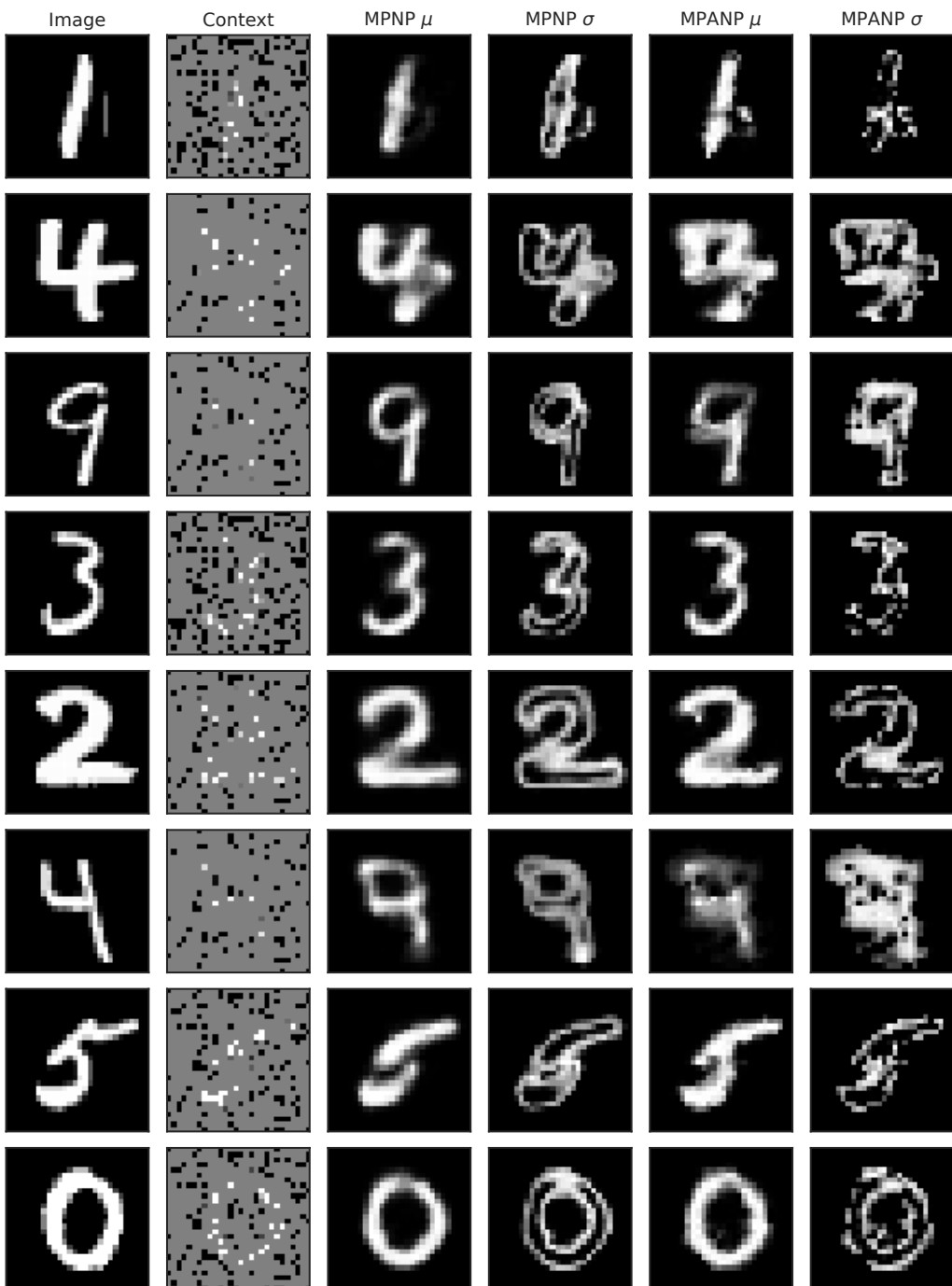

**Figure 6:** Predicted mean and standard deviation of image pixels by trained MPNPs with MNIST dataset. The first column shows the real image from test dataset. The second column shows the context dataset which given to the models. The third and the forth columns show the predicted mean and standard deviation from the MPNP respectively. The fifth and the sixth columns show the predicted mean and standard deviation from the MPANP.

**MNIST** We split MNIST (LeCun et al., 1998) train dataset into train set with 50,000 samples and validation set with 10,000 samples. We use whole 10,000 samples in test dataset as test set. We

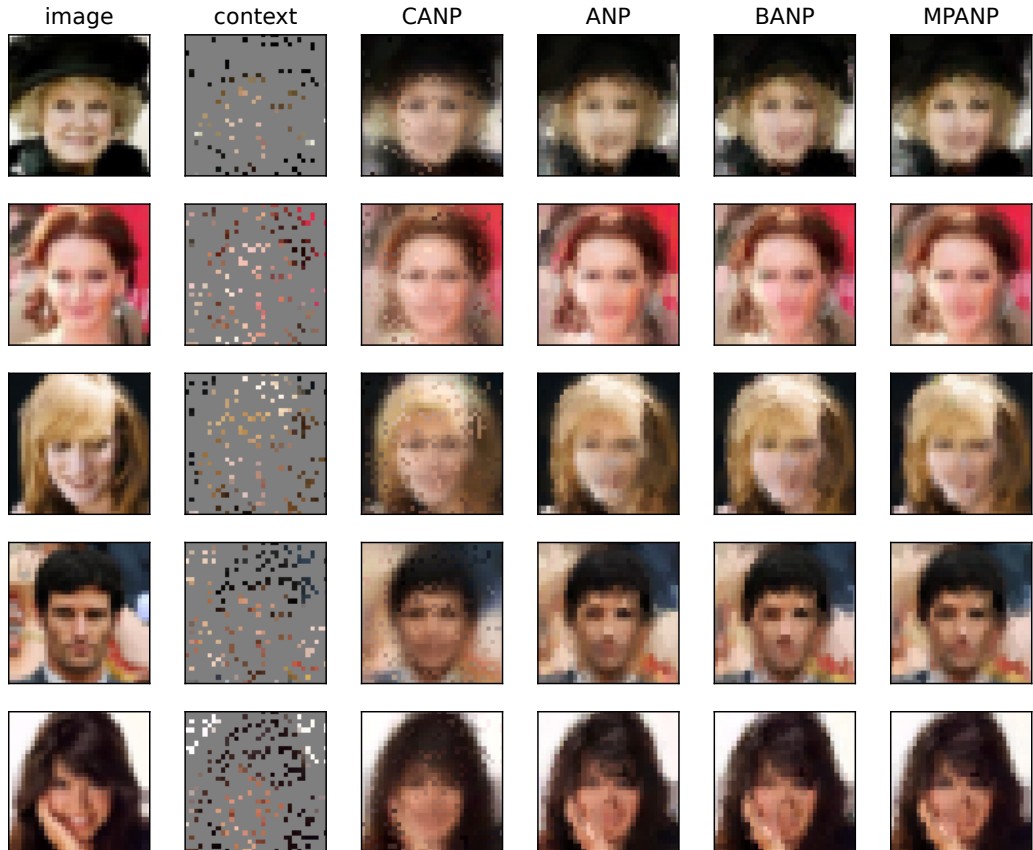

**Figure 7:** Predicted mean of image pixels by trained CANP, ANP, BANP and MPANP model. (Column 1) Here we can see the 5 ground truth real image from the test dataset. (Column 2) The context set which given to the models. (Column 3-6) The predicted mean of image pixels by each models.

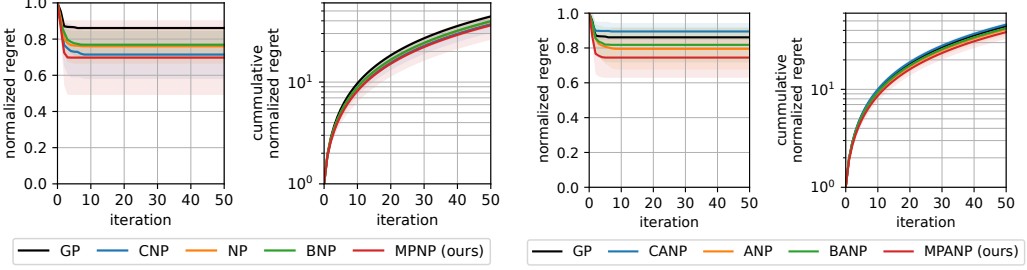

**Figure 8:** Results for Bayesian optimization on Forrester et al. (2008) function.

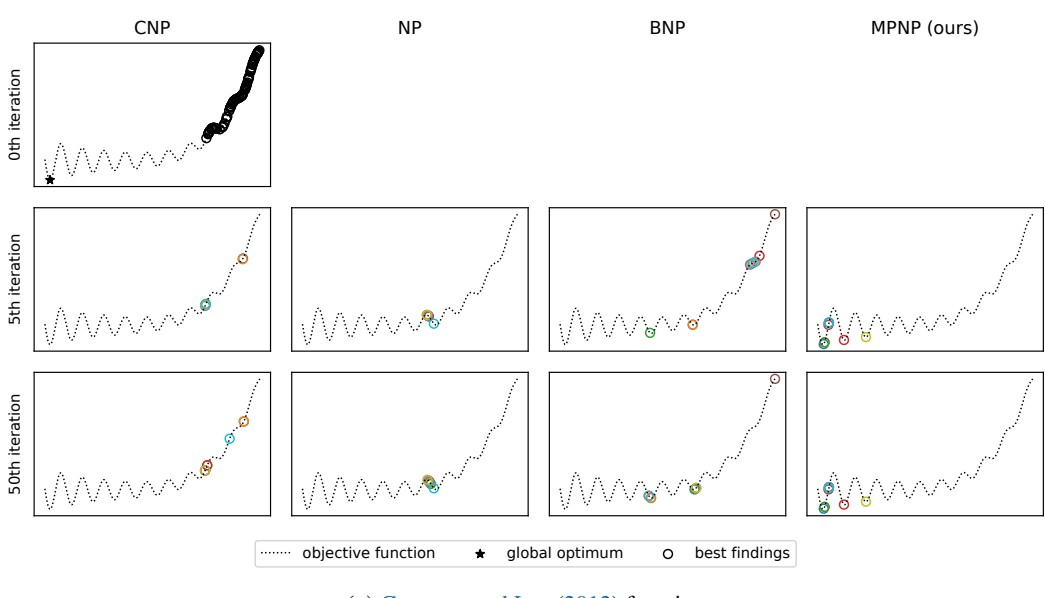

**(a)** Gramacy and Lee (2012) function

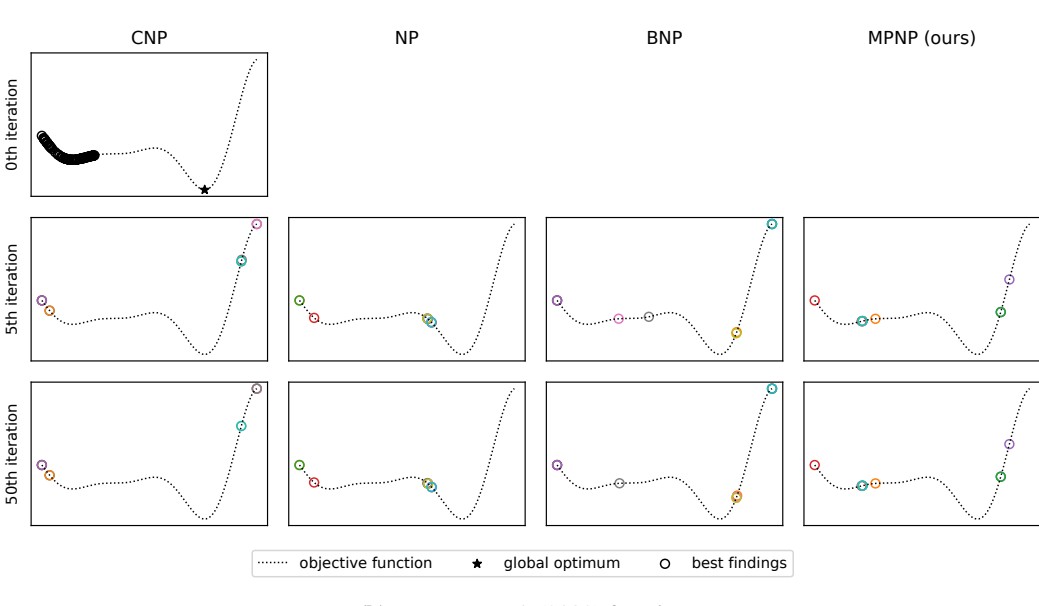

**(b)** Forrester et al. (2008) function

**Figure 9:** It depicts 10 solutions predicted by CNP, NP, BNP, and MPNP. (a,b) Predicted results for Gramacy and Lee (2012) function and Forrester et al. (2008) function, respectively. (Row 1) Black circles indicate the whole initial points. (Row 2) It shows the 10 best solutions predicted by each models after the 5 iterations. (Row 3) It shows the 10 best solutions predicted by each models after the whole iterations.

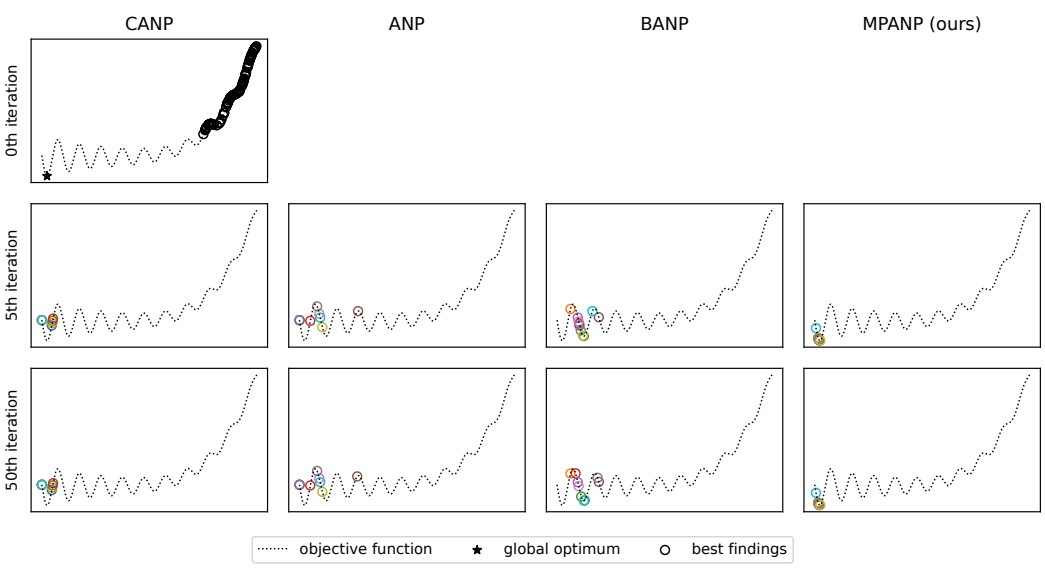

**(a)** Gramacy and Lee (2012) function

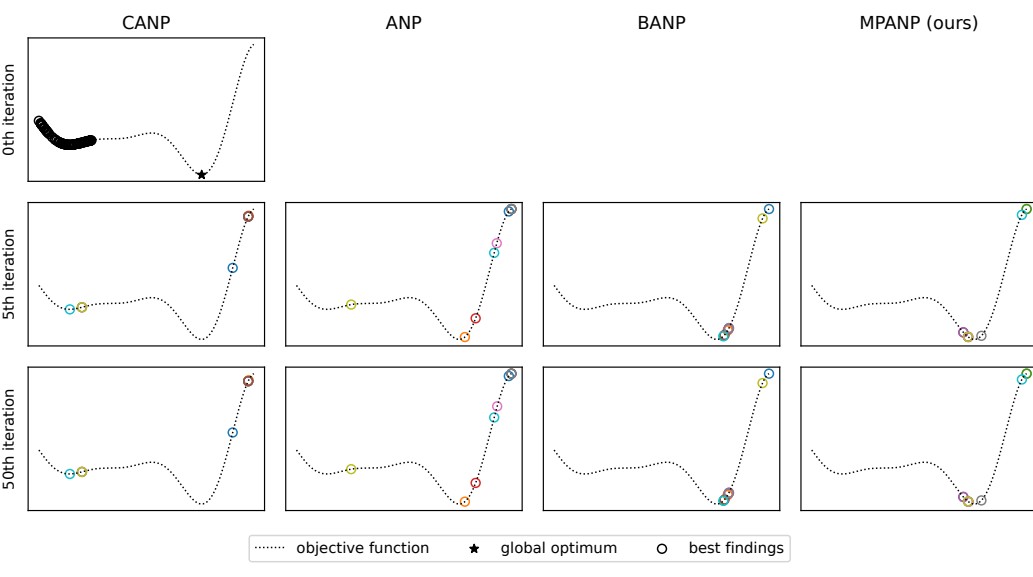

**(b)** Forrester et al. (2008) function

**Figure 10:** It depicts 10 solutions predicted by CANP, ANP, BANP, and MPANP. (a,b) Predicted results for Gramacy and Lee (2012) function and Forrester et al. (2008) function, respectively. (Row 1) Black circles indicate the whole initial points. (Row 2) It shows the 10 best solutions predicted by each models after the 5 iterations. (Row 3) It shows the 10 best solutions predicted by each models after the whole iterations.

make $28 \times 28$ grids which both axes starting from $-0.5$ to $0.5$ to indicate the coordinate of pixels, and normalize pixel values into $[-0.5, 0.5]$. We use a batch size of $128$ for training.

**SVHN**   We split SVHN (Netzer et al., 2011) train dataset into train set with 58,600 samples and validation set with 14,657 samples. We use whole 26,032 samples in test dataset as test set. We make $32 \times 32$ grids which both axes starting from $-0.5$ to $0.5$ to indicate the coordinate of pixels, and normalize pixel values into $[-0.5, 0.5]$. We use a batch size of $128$ for training.

**CelebA**   We use splits of CelebA (Liu et al., 2015) dataset as provided (162,770 train samples, 19,867 validation samples, 19,962 test samples). We crop $32 \times 32$ pixels of center of images. We make $32 \times 32$ grids which both axes starting from $-0.5$ to $0.5$ to indicate the coordinate of pixels, and normalize pixel values into $[-0.5, 0.5]$. We use a batch size of $128$ for training.

### C.4   BAYESIAN OPTIMIZATION

We use the following benchmark functions for Bayesian optimization experiments. Throughout the experiments, we adjust the function to have the domain of $[-2.0, 2.0]$.

**Gramacy and Lee (2012) function**

$$f(x) = \frac{\sin(10\pi x)}{2x} + (x - 1)^4, \tag{22}$$

where $x \in [0.5, 2.5]$ and a global optimum is at $x^* \approx 0.5486$.

**Forrester et al. (2008) function**

$$f(x) = (6x - 2)^2 \sin(12x - 4), \tag{23}$$

where $x \in [0, 1]$ and a global optimum is at $x^* \approx 0.7572$.

## D   DIRECTLY GENERATING INPUT MODEL

In this section, we present our model generating pseudo contexts directly in the input space. We will present two kinds of model structure, i) directly generating pseudo context pair $(x, y)$ simultaneously by ISAB, ii) generating pseudo context data $x$ and $y$, sequentially.

### D.1   CONSTRUCTION

**Generating pseudo context pair simultaneously.**   The generator of our first model which simultaneously generating pseudo context pair $(x', y')$, takes real context dataset $Z_c$ as input and outputs pseudo context dataset $Z'$. Here the generator is the one layer ISAB module. Then we concatenate $Z_c$ and $Z'$ in order to treat this concatenated set as context dataset. Then the encoder takes this concatenated context set as input. And the others are the same with CNP or CANP.

**Sequentially generating pseudo context data $x$ and $y$**   In this model, the generator takes real context dataset $Z_c$ as input and outputs only $x'$s of $Z'$. Here the generator is the one layer ISAB module with additional one linear layer. Then we consider these $x'$s as our target dataset and find the mean and variance of $y'$ for each $x'$ by forwarding the model with context dataset $Z_c$ and target $x'$. We sample $y'$ from the Gaussian distribution with mean and variance from the prior step. We again concatenate $Z_c$ with $Z'$ and use them as context dataset.

**Training**   Having directly generated a pseudo context set, we construct our empirical density as

$$g_N(z) = \frac{1}{N} \left( \sum_{i \in c} \delta_{z_i}(z) + \sum_{i=1}^{N-|c|} \delta_{z'_i}(z) \right). \tag{24}$$

**Table 7:** Test results for 1D regression tasks on RBF. 'Context' and 'Target' respectively denote context and target log-likelihood values, and 'Task' denotes the task log-likelihood. All values are averaged over four seeds.

| Model | RBF | | |
| --- | --- | --- | --- |
| | Context | Target | Task |
| CNP | $1.096_{\pm 0.023}$ | $0.515_{\pm 0.018}$ | $0.796_{\pm 0.020}$ |
| NP | $1.022_{\pm 0.005}$ | $0.498_{\pm 0.003}$ | $0.748_{\pm 0.004}$ |
| BNP | $1.112_{\pm 0.003}$ | $0.588_{\pm 0.004}$ | $0.841_{\pm 0.003}$ |
| **MPNP (ours)** | $\mathbf{1.189}_{\pm 0.005}$ | $\mathbf{0.675}_{\pm 0.003}$ | $\mathbf{0.911}_{\pm 0.003}$ |
| **MPNP DSI(ours)** | $1.120_{\pm 0.007}$ | $0.551_{\pm 0.006}$ | $0.822_{\pm 0.007}$ |
| **MPNP DSE(ours)** | $1.121_{\pm 0.007}$ | $0.555_{\pm 0.006}$ | $0.824_{\pm 0.007}$ |
| CANP | $1.304_{\pm 0.027}$ | $0.847_{\pm 0.005}$ | $1.036_{\pm 0.020}$ |
| ANP | $\mathbf{1.380}_{\pm 0.000}$ | $0.850_{\pm 0.007}$ | $1.090_{\pm 0.003}$ |
| BANP | $\mathbf{1.380}_{\pm 0.000}$ | $0.846_{\pm 0.001}$ | $1.088_{\pm 0.000}$ |
| **MPANP (ours)** | $1.379_{\pm 0.000}$ | $\mathbf{0.881}_{\pm 0.003}$ | $\mathbf{1.102}_{\pm 0.001}$ |
| **MPANP DSI(ours)** | $\mathbf{1.380}_{\pm 0.000}$ | $0.796_{\pm 0.013}$ | $1.069_{\pm 0.005}$ |
| **MPANP DSE(ours)** | $\mathbf{1.380}_{\pm 0.000}$ | $0.783_{\pm 0.014}$ | $1.064_{\pm 0.005}$ |

Given $g_N$, we find the function parameter $\theta$ as

$$\theta(g_N) := \arg\min_\theta \int \ell(z,\theta)g_N(dz), \tag{25}$$

where we simply choose $l(z,\theta) := -\log \mathcal{N}(y|\mu_\theta(x), \sigma_\theta^2(x)I_{d_{out}})$. In order to train the directly generating input model, which well approximate $\theta(g_N)$, we should construct different objective function from Eq. 19 because we can compute the exact $\int \ell(z,\theta)g_N(dz)$, unlike the feature generating model. First, we approximate the marginal likelihood which is,

$$\log p(Y|X, Z_c) \approx \log \left[ \frac{1}{K} \sum_{k=1}^{K} \exp\left( -\sum_{i\in[n]} \ell(z_i, \tilde{\theta}(Z_c \cup Z'^{(k)})) \right) \right] := -\mathcal{L}_{\text{marg}}(\tau, \phi), \tag{26}$$

where $Z'^{(1)}, \ldots, Z'^{(K)} \overset{\text{i.i.d.}}{\sim} p(Z'|Z_c; \phi_{\text{pred}})$. Eq. 26 is the same training object with Eq. 16. As we mentioned in Section 3.2, if we are given sufficiently well approximated $\tilde{\theta}(Z_c \cup Z'^{(K)})$ then this objective would be suffice. However only with Eq. 26, we cannot train the encoder to properly amortize the parameter construction process Eq. 11. To overcome this issue, we use $\int \ell(z,\theta)g_N(dz)$ as our second training objective which is,

$$\frac{1}{K}\sum_{k=1}^{K} \int \ell(z,\theta)g_N^{(k)}(dz) = \frac{1}{K}\sum_{k=1}^{K} \sum_{z\in Z_c\cup Z'^{(k)}} \left( -\ell(z, \tilde{\theta}(Z_c \cup Z'^{(k)})) \right) := \mathcal{L}_{\text{amort}}(\tau, \phi). \tag{27}$$

Combining these two functions, our loss function for the direct MPNP is then

$$\mathbb{E}_\tau[\mathcal{L}(\tau, \phi)] = \mathbb{E}_\tau[\mathcal{L}_{\text{marg}}(\tau, \phi) + \mathcal{L}_{\text{amort}}(\tau, \phi)]. \tag{28}$$

### D.2 Sample

In this section, we presents how the directly generating input model actually samples the pseudo context datasets.

In Fig. 11, we report generated pseudo context datasets and posterior samples from two different cases of directly generating input models for 1D regression task with RBF kernel. Here we can see that the generator samples pseudo context datasets far from the real context dataset. This phenomenon occurs because the generator learns to generate meaningless inputs ignored by the decoder. In Fig. 12, we report how two different directly generating MPANPs predict posterior samples for 1D regression task with RBF kernel. Although directly generated pseudo context dataset are a bit far from context dataset, our model still well capture the functional uncertainty in this case. We report

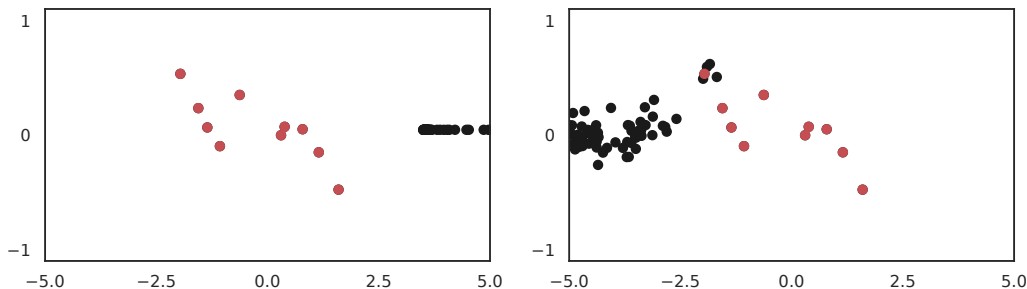

**Figure 11:** It shows generated pseudo context dataset of direct MPANP for 1D regression task with RBF kernel. The red dots are true context points sampled from GP with RBF kernel, and the black dots are generated pseudo context points. (Left) Results from simultaneously generating pseudo context pair MPANP model. (Right) Results from sequentially generating pseudo context data MPANP model.

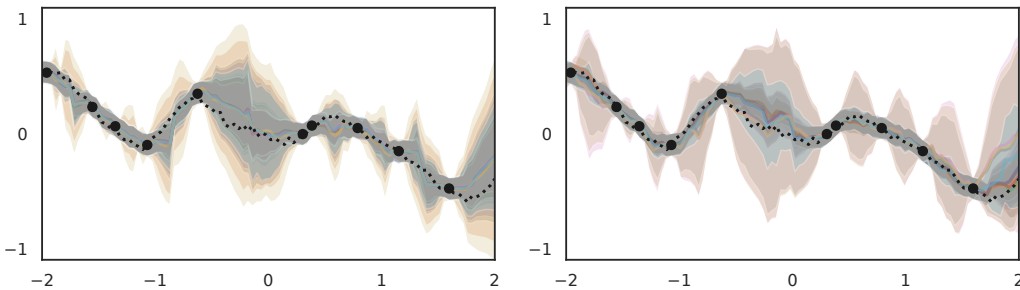

**Figure 12:** It shows posterior samples of direct MPANP for 1D regression task with RBF kernel. The black dashed line is a function sampled from GP with RBF kernel, and the black dots are context points. We visualized decoded mean and standard deviation with colored lines and areas. (Left) Results from simultaneously generating pseudo context pair MPANP model. (Right) Results from sequentially generating pseudo context data MPANP model.

the test results for 1D regression tasks on RBF for two directly generating models in Table 7. DSI and DSE indicate simultaneously generating models and sequentially generating models, respectively. Table 7 shows that our directly generating models still outperform CNP and CANP in the perspective of log-likelihood.

