# OpenReview forum: "Martingale Posterior Neural Processes"
_ICLR.cc/2023/Conference — ICLR 2023 notable top 25%_

### Official Review · Reviewer_apWK · 2022-10-15

**Confidence:** 4
**Correctness:** 4
**Technical Novelty And Significance:** 4
**Empirical Novelty And Significance:** 3
**Recommendation:** 8

**Clarity, Quality, Novelty And Reproducibility:**

The paper is written in a precise, technically correct, and easy-to-follow scientific language. It also provides sufficient details to reproduce the experiments.

The proposed idea, using martingale posterior inference for neural processes, is novel, not only as being the first combination of the two ideas but also as being a useful and meaningful combination of them.

**Strength And Weaknesses:**

Strengths:

i) The idea is overall interesting and novel. Application of the martingale posterior idea to neural processes is a rather straightforward and almost literal but anyway useful and suitable combination of two methodologies.

ii) Bayesian optimization results are strong and image completion results are OK.

Weaknesses:

i) The consistent outperformance of BNP/BANP over MPNP/MPANP weakens the central hypothesis of the paper. Why would one expect that the martingale posterior approach to outperform Bayesian inference in that particular setup and why was it not the case?

ii) The comparisons appear to be against relatively old versions of NPs. I wonder how the proposed method compares against more recent versions of NPs than ANPs (2018) and BNPs (2020), for instance Evidential Turing Processes (2022).

https://openreview.net/forum?id=84NMXTHYe-

iii) I find that the adaptation of the MPNP idea to CANP a bit dilutes the main message of the paper. It is after all a heavy pipeline with many components. The contribution of martingale posterior inference to performance would be more directly measurable when one uses a simple pipeline. It is also a bit misleading to have a Fig 1 which describes some architecture details that do not have much relevance to martingale posteriors. It could be shifted to the appendix and replaced by another Fig 1, maybe a plate diagram that gives a gist of the presented inference technique.

iv) It is great that the paper points out the limitations of the presented method, but would be even better if it also gave an educated guess on which properties of the method cause them. For instance the model-data mismatch and the decoder input ignorance. How would the authors explain these seemingly unexpected outcomes?


**Summary Of The Paper:**

The proposed method uses the martingale posterior technique in place of Bayesian inference for the first time to construct neural processes, i.e. stochastic processes generated using neural networks. Martingale posterior approach models the predictive distribution straight ahead, by passing the need for an interim approximate Bayesian inference step. The combination of two approaches, neural processes and martingale posterior inference, makes intuitive sense. Its implementation appears to be in order and some experiment results are strong.


**Summary Of The Review:**

This is solid work that presents a novel and interesting idea backed with a sufficient set of experimental results.

---

> ### Author Response · Authors · 2022-11-14
> **Response for Reviewer apWK**
>
> We thank you for your constructive and positive comments on our paper.
>
> Q: The consistent outperformance of BNP/BANP over MPNP/MPANP weakens the central hypothesis of the paper. Why would one expect that the martingale posterior approach to outperform Bayesian inference in that particular setup and why was it not the case?
>
> A: We first note that BNP/BANP only outperforms for model-data mismatch settings (sim2real experiments), where BNP/BANP are particularly designed to excel on those tasks (please refer to our answer to the Reviewer ejFI for this). Other than that, MPNP significantly outperforms BNP/BANP, since it can flexibly model the functional uncertainty through exchangeable generative models and corresponding martingale posteriors, while BNP/BANP is restricted to residual bootstrap in driving functional uncertainty.
>
> Q: The comparisons appear to be against relatively old versions of NPs. I wonder how the proposed method compares against more recent versions of NPs than ANPs (2018) and BNPs (2020), for instance Evidential Turing Processes (2022).
>
> A: Thank you for your kind suggestion to make our paper solid. However, at this moment, it is hard to make a convincing comparison to Kandemir et al. (2021) since they considered classification tasks contrary to our work dealing with regression tasks. Nevertheless, we believe our approach is not confined to regression tasks and can operate on top of general neural process models. It would be interesting for future work to investigate how we can combine ours with such classification models.
>
> Q: I find that the adaptation of the MPNP idea to CANP a bit dilutes the main message of the paper. It is after all a heavy pipeline with many components. The contribution of martingale posterior inference to performance would be more directly measurable when one uses a simple pipeline. It is also a bit misleading to have a Fig 1 which describes some architecture details that do not have much relevance to martingale posteriors. It could be shifted to the appendix and replaced by another Fig 1, maybe a plate diagram that gives a gist of the presented inference technique.
>
> A: We agree extra components irrelevant to our approach may distract from the main idea. In the revised version of our paper, Figure 1 describes the proposed pipeline built upon CNP (the prior illustration with CANP is now available in the appendix).
>
> Q: It is great that the paper points out the limitations of the presented method, but would be even better if it also gave an educated guess on which properties of the method cause them. For instance the model-data mismatch and the decoder input ignorance. How would the authors explain these seemingly unexpected outcomes?
>
> A: Thank you for the interesting question. While Martingale posteriors provide a flexible and generic extension of Bayesian posteriors, it does not particularly assume the model-data mismatch in its design, so for now it is not clear in what mechanism the martingale posterior fails for such a setting. For the decoder input ignorance problem, our rationale for it happening is,
> 1) In Fong et al., (2019), the generative model $p(z_i|z_1, \dots, z_{i-1})$ is pretrained with maximum marginal likelihood, but here, we use an implicit but exchangeable generative model from which we can draw samples but cannot evaluate the likelihoods. This choice would enhance the flexibility of the generative model, but make it harder to tame, especially in an end-to-end fashion along with the encoder and decoder.
> 2) At the early stage of the training, our generative model is likely to produce noisy data that are not really helpful for the prediction, so for the encoder and decoder it is better to ignore the pseudo-inputs generated from the generative models. Hence, an obvious way to cheat is to just generate samples that are far away from the actual task data, so that the encoder and decoder ignore them.
>
> [1] J. H. Huggins and J. W. Miller. Using bagged posteriors for robust inference and model criticism. arXiv preprint arXiv:1912.07104, 2019.
>
> [2] Kandemir, M., Akgül, A., Haussmann, M., and Unal, G. Evidential Turing Processes. In International Conference on Learning Representations. 2021.
>
> [3] E. Fong, C. Holmes, and S. G. Walker. Martingale posterior distributions. arXiv preprint arXiv:2103.15671, 2021.

---

> > ### Comment · Reviewer_apWK · 2022-11-28
> > **Keep my grade**
> >
> > Thanks for your response. My concerns have been addressed.

---

### Official Review · Reviewer_ejFi · 2022-10-24

**Confidence:** 3
**Correctness:** 4
**Technical Novelty And Significance:** 4
**Empirical Novelty And Significance:** 4
**Recommendation:** 8

**Clarity, Quality, Novelty And Reproducibility:**

## Originality

The authors explore a novel direction to increase NP's flexibility in modelling epistemic uncertainty based on the recent work of Fong et al. (2021).

## Clarity

The paper is well-written and easy to follow. As stated above, I appreciated the clear outline of background and prior work.

## Reproducibility

It seems that sufficient details for reproducibility are provided; I have not reviewed the included code.

**Strength And Weaknesses:**

## Strengths

- The paper explores an interesting new direction to account for epistemic uncertainty in NPs.
- Across a range of different tasks MPNPs show competitive empirical performance over baselines.
- The paper may open a new avenue for further improvements of NP methods.
- Background and prior work are clearly laid out.

## Weaknesses

- It would be informative to see how MPNPs scale with higher dimensionality. For example, empirical comparisons on a high-D regression task complementing the 1D one.
- The results of the Lotka-Volterra task would deserve further analysis: Why is BNP/BANP seemingly more apt at dealing with misspecification than MPNPs? My understanding is that model data-mismatch is a problem general to Bayesian inference, i.e., should also affect B(A)NP.

**Summary Of The Paper:**

The paper introduces a novel Neural Process (NP) variant which uses martingale posterior distributions to account for epistemic uncertainty. Martingale posteriors are a recent generalization of Bayesian inference proposed by Fong et al. (2021), in which valid posteriors are specified entirely in terms of the predictive distribution function (rather than relying on the traditional prior-likelihood construction). Martingale Posterior Neural Processes (MPNPs) are compared against existing NP variants on a set of different tasks, and overall outperforming the baselines.

**Summary Of The Review:**

The paper provides a novel extension of NPs with strong empirical performance based on martingale posteriors and seems like a valuable contribution to the conference.

Two possible points for improvement are 1) inclusion of higher-D tasks, and 2) further analysis of model-data misspecification.

---

> ### Author Response · Authors · 2022-11-14
> **Response for Reviewer ejFi**
>
> We thank you for your constructive and positive comments on our paper.
>
> Q: It would be informative to see how MPNPs scale with higher dimensionality.
>
> A: Following your suggestion, we conducted additional experiments on the synthetic high-dimensional regression data (i.e., generating one-dimensional y from four-dimensional x with RBF kernel). Table A1 clearly shows that our MPNPs still outperform baselines for all log-likelihood values we measured.
>
> Table A1. Test results for 4D regression tasks on RBF. ‘Context’ and ‘Target’ respectively denote
> context and target log-likelihood values, and ‘Task’ denotes the task log-likelihood. All values are
> averaged over four seeds.
> | Model        | Context     | Target      | Task        |
> | :-           | :-          | :-          | :-          |
> | CNP          | 0.572±0.003 | 0.265±0.002 | 0.410±0.003 |
> | NP           | 0.568±0.009 | 0.267±0.004 | 0.407±0.007 |
> | BNP          | 0.621±0.015 | 0.323±0.008 | 0.467±0.013 |
> | MPNP (ours)  | **0.820**±0.002 | **0.441**±0.004 | **0.633**±0.004 |
> |              |             |             |             |
> | CANP         | 0.957±0.005 | 0.585±0.006 | 0.743±0.005 |
> | ANP          | 1.357±0.006 | 0.320±0.014 | 0.890±0.007 |
> | BANP         | **1.380**±0.000 | 0.549±0.006 | 1.013±0.002 |
> | MPANP (ours) | **1.379**±0.000 | **0.645**±0.007 | **1.046**±0.002 |
>
>
> We added these results in appendix B.2.
>
> Q: Why is BNP/BANP seemingly more apt at dealing with misspecification than MPNPs?
>
> A: As first described in Huggins et al., (2019) and reconfirmed in Lee et al., (2020), bagging multiple posteriors computed from bootstraps enjoys robustness to model-data mismatch since it combines model uncertainty with additional data uncertainty coming from bootstrapping. Lee et al., (2020) also describe the mechanism of BNP/BANP adapting to mismatch data; when a BNP/BANP model sees OOD data, it will produce erroneous initial predictions, and uncertainties in those initial predictions are later encoded into bootstrapped parameters. On the other hand, MPNP is not particularly designed to handle such a mismatch scenario in its design.
>
> [1] J. H. Huggins and J. W. Miller. Using bagged posteriors for robust inference and model criticism. arXiv preprint arXiv:1912.07104, 2019.
>
> [2] J. Lee, Y. Lee, J. Kim, E. Yang, S. J. Hwang, and Y. W. Teh. Bootstrapping neural processes. In Advances in Neural Information Processing Systems 33 (NeurIPS 2020), 2020.

---

### Official Review · Reviewer_eSf2 · 2022-10-25

**Confidence:** 3
**Correctness:** 4
**Technical Novelty And Significance:** 3
**Empirical Novelty And Significance:** 3
**Recommendation:** 10

**Clarity, Quality, Novelty And Reproducibility:**

Clarity: the authors used a very specific type of expressions throughout the natural language part of this paper, and I found it difficult to understand without guessing on the way.

What does "each task" correspond to in "here each task in the data stream consists of a meta-training set of input-output pairs and also a meta-validation set"?

Why does "taking the meta-training set as an input" mean? How can a neural network take a dataset as input? I had to guess if it means training a neural net on a meta-training set.

And what does "map the meta-training set into a fixed dimensional global latent variable with a Gaussian posterior approximation" mean here? I guessed that the authors had a mental model of the operation that the neural process method is accomplishing, an abstraction perhaps used more often in functional analysis etc, but that's just not the kind of language that a machine learning researcher would typically use.

The introduction of "future data" was very abrupt. Its meaning was not explained as I've never seen such expression in the Bayesian machine learning literature. It's perhaps only clear to people who might have read Fong et al., 2021.

The entire introduction is not an "introduction". It is a summary of the paper and only made sense to me after reading the rest of this paper. E.g. the view of meta learning for this problem was introduced in section 2.2, and the concept of tasks in section 2.1, and  I think it could be much improved by writing a motivation and some high level overviews of NP and martingale posterior (and why they make sense to be combined for that motivation).

The rest of this work is mostly math expressions and very few explanations in words (I wished there were more explanations for the illustration which is supposed to be intuitive), making it difficult to understand for perhaps most machine learning researchers who specialize more on deep learning. Given that this is a machine learning conference on representation learning, I don't know how well this paper can convey its core ideas to the attendees without having the readers getting into the weed. But writing aside, I think if people read the math descriptions, they would get it.

Quality: I think it's good overall. I didn't spot critical mistakes or logical issues. The authors conducted extensive experiments on multiple tasks and presented the promise, as well as potential limitations.

The notations are well defined and consistent in the paper, except some parts of sec 3. The definitions of the problems on NP, martingale posterior and MPNP became clearer once I accepted the notations and expressions.

Some questions/comments I had while reading:
1. How can Y be iid given f in Eq 2? Each element of Y has a distribution parameterized by its corresponding x. For Eq 1 to hold, independence seems to be enough.
2. how is \sigma_\theta(x) represented in f_{dec}?
3. which sequence of variables in section 2.3 exactly is a martingale?
4. what are the weak conditions for the consistency with Bayesian posterior in sec 2.3? And are those supposed to be satisfied for the proposed method? How practical are these conditions?
5. what's the relation between conformal prediction and martingale posterior? They seem to both rely on how far Z' is from Z?
6. Are the samples from Z\cup Z' iid? If so, why so? If not, why not?
7. c was introduced as a set of indices together with Z_c and Z_t, but in sec 3.1, it's unclear what the relations are for Z', Z_c and the unmentioned Z_t. They seem to overlap but it wasn't explained. By the definition of Z', another guess I had was that Z' is somehow a perturbed version of Z_c or Z. This is super confusing. Please clarify.
8. Explain what exactly is \delta_z in sec 3.1.
9. \pi_N in sec 2.3 was defined to be the martingale posterior of theta. But in sec 3.1, how is this posterior computed exactly? It seems that the distribution of theta was implicitly represented as samples from CNPs. But it's unclear how these samples are generated (or how Z' samples can be generated to ensure the distribution is well defined).
10. Explain all variables in Fig 1. IMHO Fig 1 in the current form produces more confusion than illustration.

For the Bayesian optimization results, I think the authors should add a GP based baseline that uses classic acquisition functions such as expected improvements or GP-UCB or probability of improvements given their popularity. Note that the same problem setup applies to https://arxiv.org/abs/2109.08215 as it can make use of the tasks \tau for pre-training in a similar fashion as Eq 19, except the inner part is NLL for a GP which is straightforward to implement. Similarly, I think it's worth comparing to that pre-trained GP for the 1D regression task.

Novelty: as far as I can tell, this work is novel. It is non trivial to extend NP to the martingale posterior setting.

Reproducibility: code was provided though I didn't run it.

**Strength And Weaknesses:**

This paper is conceptually very interesting but the writing is not on par with the idea. See the section below for more details.

**Summary Of The Paper:**

This paper applied the technique of martingale posterior distributions from Fong et al., 2021 to neural processes (Garnelo et al., 2018). The key idea is to use NP's NLL as the loss function to construct the distribution of the parameter of interest, which, provided observations, has an interpretation of posterior given some assumptions from Fong et al. 2021.

**Summary Of The Review:**

I like this paper but I'm somewhat afraid that it is not going to be well received at ICLR due to its writing style. While I hope the authors could focus more on writing as a way to convey information easily (in addition to compactly), I believe this work should be accepted as it presents an interesting and solid extension of neural processes by adopting the martingale posterior interpretation of uncertainty. This introduces new opportunities in deep learning based uncertainty estimation.


=====After rebuttal=======

I've read the rebuttal and updated the score accordingly. I appreciate the clarifications and the (ongoing) effort to make this paper more readable.

---

> ### Author Response · Authors · 2022-11-14
> **Response for Reviewer eSf2 #1**
>
> We thank you for your constructive and positive comments on our paper.
>
> Q: The authors used a very specific type of expressions throughout the natural language part of this paper, and I found it difficult to understand without guessing on the way.
>
> A: We thank you for your constructive comments about the composition of our paper. Even though we could not reorganize the paper within the rebuttal period, we are considering reorganizing the paper reflecting your suggestion before camera-ready.
>
> Q: How can Y be iid given f in Eq 2? Each element of Y has a distribution parameterized by its corresponding x. For Eq 1 to hold, independence seems to be enough.
>
> A: Thank you for highlighting this typo. You are correct that independence is sufficient for Eq 1. We will clarify this.
>
> Q: How is \sigma_\theta(x) represented in f_{dec}?
>
> A: As we mentioned in equation (5), our network outputs log \sigma_\theta(x) and we transform it into \sigma_\theta^2(x).
>
> Q: Which sequence of variables in section 2.3 exactly is a martingale?
>
> A: In the martingale posterior, the sequence of predictive distributions, specifically the CDF or the density if it exists (i.e. p(z_i | z_{1:i-1}) in our notation), is the martingale when z_i ~ p(z_i | z_{1:i-1}). We will add a sentence clarifying this. For more details, please see Section 3.2 in Fong et al., (2021).
>
> Q: What are weak conditions for the consistency with Bayesian posterior in sec 2.3? And are those supposed to be satisfied for the proposed method? How practical are these conditions?
>
> A: The consistency for the Bayesian posterior is exactly Doob’s theorem, where the conditions can be found in Theorem 1 of Fong et al., (2021) or in Doob, J. L. (1949). Briefly, the main conditions are that the parameter should be identifiable and E[|theta|] < infinity. We highlight that Doob’s notion of consistency is convergence of theta(g_N) to a random variable from the posterior, which differs from the usual frequentist consistency. More details for this can also be found in Appendix C.1.1 of Fong et al., 2021
> In the martingale posterior context, the limiting empirical distribution exists almost surely from the martingale, so we do not require additional assumptions here if that is the object of interest. We then define the parameter explicitly as a functional of the limiting empirical distribution in equation (11), which is not necessarily unique so Doob’s theorem may not hold for theta. It is however possible to constrain the solution of (11) so that it is unique (e.g. by enforcing an ordering). In our context, this is not necessary as theta only enters through l(z,theta) (e.g. in equations (14)-(18)) which will have a unique minimum value even if theta is not unique.
>
> Q: What’s the relation between conformal prediction and martingale posterior? They seem to both rely on how far Z’ is from Z?
>
> A: Thank you for the interesting connection! We have not explored this yet. The main difference is that in conformal prediction, Z’ will be independent and have the same distribution (i.i.d.) as Z, whereas in the martingale posterior, Z’ will be jointly dependent and be distributed according to p(Z’ | Z). We will explore this connection in later work.
>
> Q: Are the samples from Z\cup Z’ iid? If so, why so? If not, why not?
>
> A: As we mentioned above question, because we draw Z’ jointly from p(Z’|Z), Z\cup Z’ is not i.i.d.
>
> Q: c was introduced as a set of indices together with Z_c and Z_t, but in sec 3.1, it’s unclear what the relations are for Z’, Z_c and the unmentioned Z_t. They seem to overlap but it wasn’t explained. By the definition of Z’, another guess I had was that Z’ is somehow a perturbed version of Z_c or Z. This is super confusion. Please clarify.
>
> A: Thank you for pointing out this confusion. In sec 3.1, Z’ is a generated pseudo context dataset by using Z_c (which is a real context dataset). We then input Z’ \cup Z_c, which is a union of real and pseudo context datasets, into f_{enc} instead of Z_c in order to obtain \tilde{\theta}(Z’\cup Z_c) which is a proxy for \theta(g_N). Up until this point, we do not use Z_t, which is a target dataset used for meta-validation.
>
>
> Q: Explain what exactly is \delta_z in sec 3.1.
>
> A: We add a definition of Dirac delta function in sec 2.2 where \delta first appears in our paper.

---

> > ### Author Response · Authors · 2022-11-14
> > **Response for Reviewer eSf2 #2**
> >
> > Q: \pi_N in sec 2.3 was defined to be the martingale posterior of theta. But in sec 3.1, how is this posterior computed exactly? It seems that the distribution of the theta was implicitly represented as samples from CNPs. But it’s unclear how these samples are generated (or how Z’ samples can be generated to ensure the distribution is well defined).
> >
> > A: In Fong et al. (2021), the copula method is used to sample Z’ from p(Z’|Z_c), where the parameters of the copula method are obtained by maximizing log p(z_i|z_1,...,z_{i-1}) where z_i’s are the elements of Z_c. The posterior sample of theta is then computed from Z’ \cup Z_c.
> >
> > In our method, we instead use the ISAB model (Lee et al., 2019) for our set generative model, which generates samples Z’ conditional on Z_c. As our generative model is exchangeable, the limiting empirical distribution exists, so the posterior over g_N exists. A posterior sample of theta can then be computed from Z’ \cup Z_c (i.e. g_N) through equation (11). On difference in our setting is that we cannot directly calculate log p(z_i|z_1,...,z_{i-1}). To overcome this, we suggest an additional training loss in section 3.2, where L_{pseudo} maximizes log p(Z|Z’) where Z=Z_c\cup Z_t.
> >
> > Q: Explain all variables in Fig 1. IMHO Fig 1 in the current form produces more confusion than illustration.
> >
> > A: Thank you for pointing out this confusion. In order to reduce confusion, we changed our illustration from MPANP to MPNP which has a simpler structure.
> >
> > Q: For the Bayesian optimization results, I think the authors should add a GP based baseline.
> >
> > A: Following your suggestion, we have added a GP based baseline tuned by following Wang et al., 2022 in the revised section 5.3. We used the RBF kernel and trained with the expected improvements acquisition function. Here we can see that the GP baseline outperforms compared to NP variants but underperforms compared to ANP variants.
> >
> > [1] Doob, J. L. (1949). Application of the theory of martingales. Actes du Colloque International Le Calcul des Probabilites et ses applications (Lyon, 28 Juin–3 Juillet 1948), Paris CNRS, 23–27.
> >
> > [2] E. Fong, C. Holmes, and S. G. Walker. Martingale posterior distributions. arXiv preprint arXiv:2103.15671, 2021.
> >
> > [3] J. Lee, Y. Lee, J. Kim, A. Kosiorek, S. Choi, and Y. W. Teh. Set transformer: A framework for attention-based permutation-invariant neural networks. In Proceedings of The 36th International Conference on Machine Learning (ICML 2019), 2019.
> >
> > [4] Z. Wang, G. E. Dahl, K. Swersky, C. Lee, Z. Mariet, Z. Nado, J. Gilmer, J. Snoek, and Z. Ghahramani. Pre-trained gaussian processes for bayesian optimization. arXiv preprint arXiv:2109.08215, 2022.

---

### Decision · Program_Chairs · 2023-01-20

**Decision:**

Accept: notable-top-25%

**Justification For Why Not Higher Score:**

I think this could be an oral presentation.  It seems technically strong and novel, and producing stochastic processes with good uncertainty using neural networks is a relatively new and exciting direction.  One reviewer gave a 10 while two stayed at 8.  One of the  reviewers found that the comparisons were to older variants of neural processes and could have had stronger baselines.  I'd say the paper could be a bit technically niche for the general ICLR community.

**Justification For Why Not Lower Score:**

The paper is technically strong and novel, and producing stochastic processes with good uncertainty using neural networks is a relatively new and exciting direction.

**Metareview: Summary, Strengths And Weaknesses:**

This paper develops a novel method for producing predictive uncertainty in neural processes using a Martingale posterior formulation.  This is validated empirically, comparing to existing neural process variants, on a variety of experiments including regression tasks, Bayesian optimization and image completion.  The reviewers all found the proposed approach compelling, technically strong, interesting and novel.  In general, they found the empirical evaluation compelling as well.  As a result, the reviewers were all in agreement on acceptance (10, 8, 8).  The Martingale posterior approach seems well suited to the problem and the reviewers felt that it could provide exciting avenues for future work.  Therefore the recommendation is to accept.

**Note From Pc:**

if the above contains the word "oral" or "spotlight" please see: "oral" presentation means -> notable-top-5% and "spotlight" means -> notable-top-25%. As stated in our emails, we are disassociating presentation type from AC recommendations